# Auto-RAG: Autonomous Retrieval-Augmented Generation for Large Language Models

## Abstract

Iterative retrieval refers to the process in which the model continuously queries the retriever during generation to enhance the relevance of the retrieved knowledge, thereby improving the performance of Retrieval-Augmented Generation (RAG). Existing work typically employs few-shot prompting or manually constructed rules to implement iterative retrieval. This introduces additional inference overhead and overlooks the remarkable reasoning capabilities of Large Language Models (LLMs). In this paper, we introduce **Auto-RAG**, an autonomous iterative retrieval model centered on the LLM's powerful decision-making capabilities. Auto-RAG engages in multi-turn dialogues with the retriever, systematically planning retrievals and refining queries to acquire valuable knowledge. This process continues until sufficient external information is gathered, at which point the results are presented to the user. To this end, we develop a method for autonomously synthesizing reasoning-based decision-making instructions in iterative retrieval and fine-tuned the latest open-source LLMs. The experimental results indicate that Auto-RAG is capable of autonomous iterative interaction with the retriever, effectively leveraging the remarkable reasoning and decision-making abilities of LLMs, which lead to outstanding performance across six benchmarks. Further analysis reveals that Auto-RAG can autonomously adjust the number of iterations based on the difficulty of the questions and the utility of the retrieved knowledge, without requiring any human intervention. Moreover, Auto-RAG expresses the iterative retrieval process in natural language, enhancing interpretability while providing users with a more intuitive experience.

## 1 Introduction

Retrieval-augmented generation (RAG) for Large Language Models (LLMs) is widely employed to tackle knowledge-intensive tasks (Asai et al., 2023; Dubey et al., 2024; Jiang et al., 2023; Feng et al., 2023; Gao et al., 2024), which substantially improves output quality and effectively mitigates hallucinations (Gao et al., 2024; Lewis et al., 2020). However, certain limitations persist, such as noise in retrieved content (Yu et al., 2023) and the challenge of retrieving sufficient knowledge for complex queries in a single attempt (Feng et al., 2023; Chen et al., 2024). These issues ultimately undermine the overall performance of RAG systems and impede their widespread adoption.

To address these limitations, iterative retrieval has been proposed, which consistently updates retrieval results to satisfy the dynamic information needs that arise during the generation process (Feng et al., 2023; Chen et al., 2024; Asai et al., 2023). Existing work often relies on few-shot prompting and manually crafted rules to implement iterative retrieval (Jiang et al., 2023; Feng et al., 2023; Wang et al., 2024a), which involves substantial human effort and additional computational overhead during inference. Moreover, these methods overlook LLMs' reasoning and decision-making capabilities (Wei et al., 2023), wasting their potential on determining when and what to retrieve.

To this end, we introduce **Auto-RAG**, an autonomous iterative retrieval model centered on the LLM's powerful decision-making capabilities. As shown in Figure 1, Auto-RAG models the interaction between the LLM and the retriever through multi-turn dialogue. During iterative retrieval, Auto-RAG employs reasoning for retrieval planning, extracting valuable external knowledge, identifying information needs, rewriting queries, and continuously querying the retriever for new information until it can adequately answer the user's question. To empower LLMs with the ability

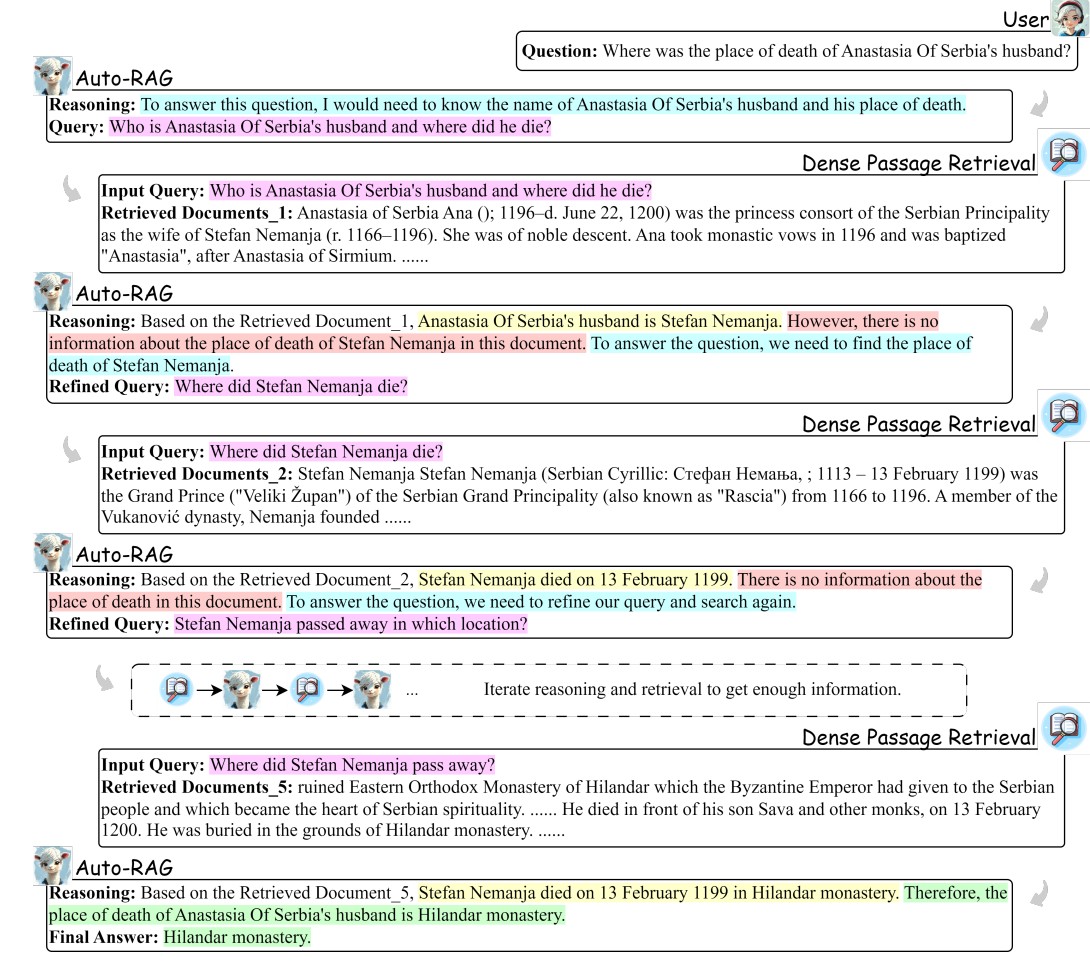

Figure 1: A concrete example of how Auto-RAG addresses complex multi-hop questions. Auto-RAG engages in iterative reasoning, strategically plans retrievals, extracts relevant knowledge, precisely identifies information needs, and refines query for the next retrieval, ultimately converging on the final answer. In this example, Auto-RAG terminates after five interactions with the retriever, successfully yielding the correct answer.

for autonomous decision-making in iterative retrieval, we developed a framework for the automatic synthesis of reasoning-based instructions for autonomous decision-making in iterative retrieval and fine-tuned the latest open-source LLMs, such as Llama-3-8B-Instruct [1] (Dubey et al., 2024).

We conduct experiments on six representative benchmarks, covering both open-domain QA (Kwiatkowski et al., 2019; Joshi et al., 2017; Berant et al., 2013; Mallen et al., 2023) and multi-hop QA (Ho et al., 2020; Yang et al., 2018). Experimental results demonstrate that, even with limited training data, Auto-RAG delivers outstanding performance. Further analysis reveals that Auto-RAG dynamically adjusts the number of iterations based on the complexity of the questions and the relevance of the retrieved knowledge. Moreover, Auto-RAG expresses the iterative retrieval process in natural language, thereby improving interpretability and offering a more intuitive user experience.

## 2 RELATED WORK

**Retrieval-Augmented Generation (RAG)** To address the challenges of outdated knowledge embedded in model parameters (Zhao et al., 2024) and the inadequate retention of long-tail knowl-

---

[1] https://huggingface.co/meta-llama/Meta-Llama-3-8B-Instruct

edge by LLMs (Mallen et al., 2023), Retrieval-Augmented Generation (RAG) has been introduced (Lewis et al., 2020; Chu et al., 2024; Yan et al., 2024). The most common RAG approach follows the Retrieve-Read framework (Gao et al., 2024), where retrieved documents are concatenated with the user's input to provide LLMs with external knowledge. However, retrievers are not without flaws (Gao et al., 2024), and the retrieved content may contain noise, which has been shown to degrade the RAG system's performance (Yu et al., 2024; 2023; Yoran et al., 2023; Hong et al., 2024). Recent studies have sought to improve RAG by refining query formulation (Ma et al., 2023), enhancing retrievers (Karpukhin et al., 2020; Chen et al., 2023), improving generators (Yoran et al., 2023; Yu et al., 2023), and optimizing post-processing of retrieved documents (Yu et al., 2024; Xu et al., 2023). Nonetheless, these methods overlook the growing difficulty of obtaining sufficient knowledge from a single retrieval attempt as the complexity of tasks increases (Jiang et al., 2023).

**Iterative Retrieval** Iterative retrieval was introduced to address the evolving knowledge requirements that arise when solving complex problems (Feng et al., 2023; Shao et al., 2023; Jiang et al., 2023; Trivedi et al., 2023). The core principle of iterative retrieval is determining *when and what to retrieve* (Jiang et al., 2023). For instance, ITER-RETGEN (Shao et al., 2023) concatenates the input question with the generated output from the previous iteration to form a new query for the next. While this method has achieved some success, it merely reflects existing knowledge without explicitly indicating the LLM's information needs. To address this shortcoming, FLARE (Jiang et al., 2023) uses the next generated sentence as a query, refining the previous sentence based on the retrieval results. Although this method more precisely identifies the LLM's information needs, its efficacy heavily depends on meticulously crafted few-shot prompts (Brown et al., 2020) and requires continuous retrieval and refinement, leading to substantial manual effort and increased inference costs. Self-RAG (Asai et al., 2023) trains LLMs to reflect on both retrieved and generated content. However, Self-RAG only learns to mechanically predict reflection tokens during training, without cultivating reasoning abilities, which further limits the effectiveness of this approach.

In contrast to the methods mentioned above, Auto-RAG fully releases the LLMs' potential for reasoning-based autonomous decision-making in the iterative retrieval process. Auto-RAG enables LLMs to autonomously decide when to retrieve and what to retrieve through reasoning. Compared to other iterative retrieval methods, Auto-RAG delivers superior performance and higher efficiency.

## 3 METHOD

To empower LLMs with autonomous decision-making capabilities in iterative retrieval at a minimal cost (Li et al., 2024; Chan et al., 2024), we develop a method for autonomously synthesizing reasoning-based decision-making instructions in iterative retrieval and fine-tuned the latest open-source LLMs. The following subsections will delve into the data construction processes, the training procedures, and the methodologies employed during inference.

### 3.1 REASONING-BASED ITERATIVE RETRIEVAL

We conceptualize the iterative retrieval process as a multi-turn interaction between LLM and retriever. The user's query initiates a sequence of interactions between the LLM and the retriever, continuing until sufficient knowledge is acquired to generate a final answer. In each iteration, Auto-RAG engages in meticulous reasoning based on the current state to ascertain whether additional retrieval is required and what specific information to seek. Once sufficient information is acquired, Auto-RAG ceases to generate new queries and delivers a final answer to the user.

We begin by formally delineating the objectives for reasoning-based instruction synthesis. For each input-output pair $(X, Y)$ in the original dataset $\mathcal{D}$, our goal is to curate instruction data collection, $\mathcal{D}^{\text{Inst}}$, that empowers LLMs to engage in reasoning and query refinement during iterative retrieval, ultimately converging on the correct answer, which can be formally expressed as follows:

$$(X, Y) \rightarrow [X, R_0, (Q_t, D_t, R_t)_{1 \leq i \leq T}, A], \tag{1}$$

where $T$ is the maximum iteration[2], $R_0$ denotes the reasoning performed when only the user's input $X$ is present. At the $t$-th iteration ($1 \leq t \leq T$), if the previous iteration's reasoning $R_{t-1}$ includes

---

[2]During synthesis training, $T$ is set to 10 for 2WikiMultihopQA and 5 for Natural Questions.

---

**Algorithm 1** Data Construction for Training Auto-RAG

---

**Input:** Dataset $\mathcal{D}$, Language model $\mathcal{M}$, Retriever $\mathcal{R}$, Maximum number of iterations $T$
**Output:** Iterative retrieval instruction-tuning dataset $\mathcal{D}^{\text{Inst}}$

1: Initialize a list $\mathcal{D}^{\text{Inst}}$ to store the generated data
2: **for** each input-output pair $(X, Y)$ in $\mathcal{D}$ **do**
3:      $\mathcal{M}$ predicts $R_0$ given $X$                                                      ▷ Planning
4:      $t = 1$
5:      **while** $t \leq T$ **do**                                                                ▷ At most $T$ iterations
6:          $\mathcal{M}$ generates queries $Q_{gen}$ given $X$ and $R_{t-1}$                        ▷ Sample queries
7:          $Q_t = \text{None}, D_t = \text{None}$
8:          **for** $q$ in $Q_{gen}$ **do**:
9:              $R$ retrieves documents $d$ for $q$
10:             **if** $d$ contains a sub answer of $X$ **then**
11:                 $Q_t = q, D_t = d$, **Break**
12:         **if** $Q_t$ and $D_t$ are None **then**
13:             Select a random $q$ from $Q_{gen}$ as $Q_t$
14:             Retrieve documents d for q as $D_t$
15:         $\mathcal{M}$ generates $R_t$ given $X, R_0, (Q_i, D_i, R_i)_{1 \leq i < t}, Q_t, D_t$      ▷ Reasoning and planning
16:         **if** no information need in $R_t$ **then**
17:             **Break**
18:         $t = t + 1$
19:     $M$ predicts final answer $A$ given $X, R_0, (Q_i, D_i, R_i)_{1 \leq i \leq t}$
20:     **if** $A == Y$ **then**                                                                  ▷ Filtering
21:         Append $[X, R_0, (Q_i, D_i, R_i)_{1 \leq i \leq t}, A]$ to $\mathcal{D}^{\text{Inst}}$

**Return:** $\mathcal{D}^{\text{Inst}}$

---

an information need[3], the query $Q_t$ will be sampled, and the retriever will provide the document $D_t$ for $Q_t$. The model will then generate the reasoning $R_t$ for that iteration. If the previous reasoning $R_{t-1}$ does not include an information need, the model is prompted to generate the final answer $A$.

Next, we will provide the details of how LLM is guided to perform such reasoning and query refinement. Additionally, we will elucidate the methods utilized for data filtering and formatting.

### 3.1.1 REASONING BASED PLANNING AND QUERY REFINEMENT

To optimize efficiency and ensure coherence during iterative processes, it is essential to develop a well-designed reasoning paradigm. Specifically, mirroring the human cognitive process during retrieval, we propose that iterative retrieval should incorporate three distinct types of reasoning: (1) Retrieval Planning, (2) Information Extraction, and (3) Answer Inference.

- **(1) Retrieval Planning** Upon receiving the user's question, the LLM should explicitly identify the knowledge necessary to address the query. Furthermore, upon receiving retrieved documents, the LLM must evaluate whether further retrievals are needed and, if so, specify the precise information to be sought next. Maintaining strategic planning throughout the retrieval process is crucial for improving efficiency and mitigating the risk of losing direction midway (Wang et al., 2024a).

- **(2) Information Extraction** Upon receiving retrieved documents, the LLM should adeptly extract relevant information essential for addressing the problem at hand. This human-like summarization process bolsters the LLM's capacity to filter out irrelevant information, thereby enhancing both its efficiency and accuracy in processing external knowledge(Wei et al., 2023; Xu et al., 2024).

- **(3) Answer inference** Once LLM has gathered all pertinent knowledge required to address the question, it should employ reasoning to formulate the final answer. This process enhances LLM's ability to generate accurate responses based on available information, thereby mitigating the risk of generating hallucinations (Wei et al., 2023).

These three types of reasoning collectively constitute the Chain-of-Thought utilized during iterative retrieval. To elicit such a reasoning process, we utilize few-shot prompting following Jiang et al. (2023); Brown et al. (2020); Wei et al. (2023). It is noteworthy that steps (2) and (3) are typically

---

[3]We predefined terms like "however," "no information," "find," and "refine" to signal the model's information needs. If any appear in the output, they indicate an information need.

omitted upon the initial reception of the user's question. Furthermore, if the retrieved information is found to be entirely irrelevant, step (2) is also excluded. Such adjustments enable LLMs to make informed judgments based on the actual context, rather than merely imitating demonstrations and generating hallucinations. The prompt used to elicit reasoning is presented in Appendix C.1.

With an appropriate reasoning process, LLM can iteratively refine the query based on the user input and previous retrieval plan, continually adapting to new information requirements. To generate a sufficiently diverse set of queries without being constrained by the query styles present in few-shot prompts, we utilize a more flexible prompting methodology, as shown in Appendix C.5.

### 3.1.2 DATA FILTERING AND FORMATTING

**Data filtering** The preceding subsections have thoroughly elucidated the methodologies for eliciting reasoning and query refinement in iterative retrieval. Nevertheless, there remains the possibility of reasoning artifacts or suboptimal query quality. Following Yoran et al. (2023); Asai et al. (2023), we undertake filtering for the generated reasoning and queries. In multi-hop question-answering datasets that encompass sub-answers, multiple queries are sampled at each retrieval iteration (Yoran et al., 2023; Ho et al., 2020). Each query is employed to perform the retrieval, and those queries for which the retrieved documents contain a sub-answer are retained. Moreover, to ensure the quality of the entire iterative retrieval process and the coherence of the output answers, data is retained if the final answer $A$ aligns with the reference answer $Y$ provided in the dataset. For greater clarity, we outline the framework of instruction synthesis and filtering in Algorithm 1.

**Data formatting** We conceptualize the iterative retrieval process as a multi-turn interactive dialogue. At each iteration, the user's question or retrieved documents serve as inputs, and the LLM's reasoning, retrieval planning, or final answer constitutes the output. We assume each instance in $\mathcal{D}^{\text{Inst}}$ comprises $T + 1$ iterations, where $T$ varies according to the instance. Specifically, at the 0-th iteration, the user's input $X$ forms the input instruction $x_0$, while the LLM-generated planning $R_0$, and the query used for the next iteration $Q_1$, serve as the output $y_0$. At $t$-th iteration (where $1 \leq t < T$), retrieved documents $D_t$ serve as the input $x_t$, while the LLM-generated reasoning $R_t$ and query $Q_{t+1}$ serve as the output $y_t$. Finally, at $T$-th iteration, $D_T$ serves as $x_T$, while $R_T$ and the final answer $A$ serves as $y_T$. The construction process can be expressed by the following formula:

$$x_t = \begin{cases} X & \text{if } t = 0 \\ D_t & \text{if } 0 < t \leq T \end{cases}, \; y_t = \begin{cases} \text{Concat}(R_t, Q_{t+1}) & \text{if } 0 \leq t < T \\ \text{Concat}(R_t, A) & \text{if } t = T \end{cases}. \tag{2}$$

### 3.2 TRAINING

To equip an arbitrary LLM with the capability for autonomous decision-making in iterative retrieval, we adopted a standard supervised fine-tuning strategy following Yoran et al. (2023); Jiang et al. (2024). For each instance containing $(x_t, y_t)_{0 \leq t \leq T}$, the cross-entropy loss $\mathcal{L}$ can be calculated as:

$$\mathcal{L} = - \sum_{0 \leq t \leq T} \log \Pr(y_t | x_{\leq t}, y_{<t}), \tag{3}$$

where $y_t$ denotes the output at iteration $t$, $x_{\leq t}$ represents the input up to the current iteration, and $y_{<t}$ signifies the outputs from all preceding steps.

### 3.3 INFERENCE

After training, Auto-RAG has acquired the ability to make reasoning-based autonomous decisions during iterative retrieval, effectively discerning both when and what to retrieve. During each iteration, it suffices to provide Auto-RAG with input—whether user inquiries or retrieved documents—and to extract the planned actions designated by Auto-RAG for subsequent steps. Specifically, in the 0-th iteration, Auto-RAG receives the user's question as input and subsequently generates the reasoning and planning output $y_t$. In the $t$-th iteration, if the output from the previous iteration $y_{t-1}$ includes a query $q$, this query is utilized for retrieval, and the retrieved documents $d_t$ are then provided to Auto-RAG as input, resulting in the output for that iteration $y_t$. Conversely, if the output from the previous iteration $y_{t-1}$ does not contain a query but instead presents a final answer, the iteration is concluded, and the final answer is returned to the user.

---

**Algorithm 2** Inference for Auto-RAG

---

**Input:** User input $X$, Language model $\mathcal{M}$, Retriever $\mathcal{R}$, Maximum iteration number of retrieval $T$, Maximum iteration number to request parametric knowledge $T^{PK}$

**Output:** Answer $A$ corresponding to $X$

1: $\mathcal{M}$ predicts $y_0$ given $X$
2: $t = 1$
3: **for** $1 \leq t \leq T$ **do**                          ▷ Aquiring for external knowledge
4:     **if** $y_{t-1}$ contains a query $q$ **then**
5:        $\mathcal{R}$ retrieves documents $d_t$ for $q$
6:        $\mathcal{M}$ predicts $y_t$ given $X$, $y_{<t}$ and $d_{\leq t}$
7:        $t = t + 1$
8:     **else if** $y_{t-1}$ contains a final answer $A$ **then**
9:        **Return:** $A$
10: **for** $T < t \leq T^{PK}$ **do**                   ▷ Aquiring for parametric knowledge
11:     **if** $y_{t-1}$ contains a query $q$ **then**
12:        $\mathcal{M}$ generates a document $d_t$ for $q$
13:        $\mathcal{M}$ predicts $y_t$ given $X$, $y_{<t}$ and $d_{\leq t}$
14:        $t = t + 1$
15:     **else if** $y_{t-1}$ contains a final answer $A$ **then**
16:        **Return:** $A$
17: $\mathcal{M}$ directly predicts answer $A$ for $X$
18: **Return:** $A$

---

**Utilization of parametric knowledge** Due to the limitations of the retriever and the retrieval corpus, Auto-RAG may fail to acquire the necessary knowledge to answer a question, resulting in perpetual iterations. Furthermore, the parametric knowledge of the LLM may not be effectively utilized during this process. To address this issue, we attempted to provide Auto-RAG with self-generated documents or answers. If Auto-RAG has not terminated after interacting with the retriever for $T$ iterations, the generated query is used to prompt itself to create a document, which is subsequently utilized as input for the next iteration. If Auto-RAG continues without termination after an additional $T^{PK}$ iterations, we follow Wang et al., 2024a to provide the answer produced by Auto-RAG without retrieval to the user. The prompt used to elicit parametric knowledge is shown in Appendix C.4, the pseudocode representing the inference process is presented in Algorithm 2, and examples of the synthesized instructions can be found in Appendix C.6. The experiments investigating the order of external and parametric knowledge can be found in Appendix A.3.

## 4 EXPERIMENTS

### 4.1 EXPERIMENTAL SETUP

In this paper, we focus on utilizing Auto-RAG to address question-answering (QA) tasks, encompassing both open-domain QA (Kwiatkowski et al., 2019; Joshi et al., 2017; Mallen et al., 2023; Berant et al., 2013) and multi-hop QA (Yang et al., 2018; Ho et al., 2020). To train Auto-RAG, we synthesized 10,000 reasoning-based instructions derived from two representative datasets: Natural Questions (NQ) (Kwiatkowski et al., 2019) and 2WikiMultihopQA (2Wiki) (Ho et al., 2020). We employed Llama-3-8B-Instruct[4] (Dubey et al., 2024) to synthesize the reasoning process and utilized Qwen1.5-32B-Chat[5] (Bai et al., 2023) for crafting the rewritten queries. Subsequently, we fine-tuned Llama-3-8B-Instruct using the synthe-

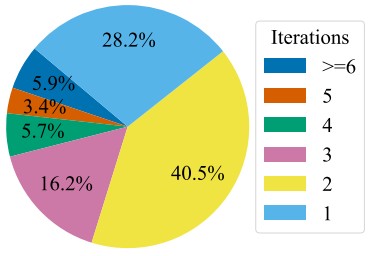

Figure 2: Distribution of iteration counts in the training data.

sized instructions for five epochs to enhance its capacity for autonomous decision-making during iterative retrieval. The distribution of iteration counts in the training data is illustrated in Figure 2. To evaluate the effectiveness and robustness of Auto-RAG, we conducted assessments across six datasets: NQ, 2Wiki, TriviaQA (TQA) (Joshi et al., 2017), PopQA (PQA) (Mallen et al., 2023),

---

[4] https://huggingface.co/meta-llama/Meta-Llama-3-8B-Instruct
[5] https://huggingface.co/Qwen/Qwen1.5-32B-Chat

Table 1: Main results on six benchmarks. Auto-RAG consistently outperforms all baselines.

| Methods | NQ | 2Wiki | TQA | PQA | HQA | WQ | AVG |
|---|---|---|---|---|---|---|---|
| | EM | F1 | EM | F1 | F1 | EM | |
| *No Retrieval* | | | | | | | |
| Naive Gen | 22.6 | 33.9 | 55.7 | 21.7 | 28.4 | 18.8 | 30.2 |
| *Single-time Retrieval* | | | | | | | |
| Standard RAG | 35.1 | 21.0 | 58.8 | 36.7 | 35.3 | 15.7 | 33.8 |
| IRCoT | 33.3 | 32.4 | 56.9 | 45.6 | 41.5 | 20.7 | 38.4 |
| REPLUG | 28.9 | 21.1 | 57.7 | 27.8 | 31.2 | 20.2 | 31.2 |
| RECOMP-abstractive | 33.1 | 32.4 | 56.4 | 39.9 | 37.5 | 20.2 | 36.6 |
| Selective-Context | 30.5 | 18.5 | 55.6 | 33.5 | 34.4 | 17.3 | 31.6 |
| *Iterative Retrieval* | | | | | | | |
| FLARE | 22.5 | 33.9 | 55.8 | 20.7 | 28.0 | 20.2 | 30.2 |
| Self-RAG | 36.4 | 25.1 | 38.2 | 32.7 | 29.6 | 21.9 | 30.7 |
| Iter-RetGen | 36.8 | 21.6 | 60.1 | 37.9 | 38.3 | 18.2 | 35.5 |
| *Ours (Autonomous Retrieval)* | | | | | | | |
| Auto-RAG | **37.9** | **48.9** | **60.9** | **47.8** | **44.9** | **25.1** | **44.3** |

HotpotQA (HQA) (Yang et al., 2018), and WebQuestions (WQ) (Berant et al., 2013). We employed E5-base-v2 (Wang et al., 2024b) as the retriever and utilized the widely used Wikipedia dump from December 2018 as the retrieval corpus (Karpukhin et al., 2020) following Jin et al. (2024). Given the variations in base models, retrievers, and retrieval corpora employed by different RAG methods, performing a fair comparison becomes challenging. Therefore, consistent with Jin et al. (2024), we report results and metrics based on their reproduction under an identical experimental setup. We present Exact Match (EM) for NQ, TQA, and WQ, and F1 scores for 2Wiki, PQA, and HQA, in accordance with Jin et al. (2024). Hyperparameters are detailed in Appendix B.

## 4.2 BASELINES

For baselines without retrieval (Naive Gen), we evaluated the performance of Llama-3-8B-Instruct. Following Jin et al. (2024), we adopted a zero-shot setting. We consider Standard RAG for retrieval-based baselines, where models generate answers based on documents retrieved by the user's input. The prompts used for Naive and Standard RAG are shown in Appendix C.2. For single time retrieval, we compare with RECOMP-abstractive (Xu et al., 2023) and Selective-Context (Li et al., 2023), which optimize on context selection, REPLUG (Shi et al., 2024), which enhances the generator's performance, and IRCoT (Trivedi et al., 2023), which adopts a Chain-of-Thought (CoT) process when reading and interpreting the retrieved documents. For multiple-time retrieval (iterative retrieval), we compare Auto-RAG with three methods that are most relevant to our approach: FLARE (Jiang et al., 2023), Iter-RetGen (Feng et al., 2023), and Self-RAG (Asai et al., 2023).

## 4.3 MAIN RESULTS

Table 1 shows the main results across six benchmarks, demonstrating that Auto-RAG achieves superior performance across all datasets. Notably, Auto-RAG surpasses other iterative retrieval methods, yielding significantly improved outcomes. While Iter-RetGen (Feng et al., 2023) relies on manually defined retrieval content and the number of iterations, and FLARE (Jiang et al., 2023) determines retrieval timing through predefined rules (e.g., output probabilities), Auto-RAG distinguishes itself by autonomously determining both when and what to retrieve, leading to superior overall performance. Self-RAG (Asai et al., 2023) directly predicts reflection tokens to decide when to retrieve and evaluate the quality of the retrieved results. In contrast, Auto-RAG incorporates a reasoning process at each iteration, enabling it to make more sophisticated and informed decisions. This reasoning mechanism enhances the Auto-RAG's capacity to optimize retrieval strategies and autonomously navigate complex tasks, resulting in improved performance across six benchmarks. Since variations in base LLMs and different versions of Wikipedia can impact performance (Izacard et al., 2022), to facilitate comparisons in future research, the results from other base models (such as the Llama-3.1-8B-Instruct Dubey et al., 2024) and different Wikipedia versions are provided in Appendix A.1. Examples of outputs generated by Auto-RAG can be found in Appendix C.7.

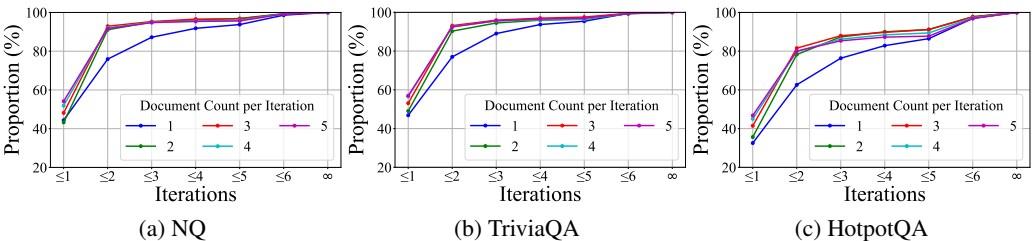

Figure 3: Auto-RAG's iteration counts across different document numbers per iteration.

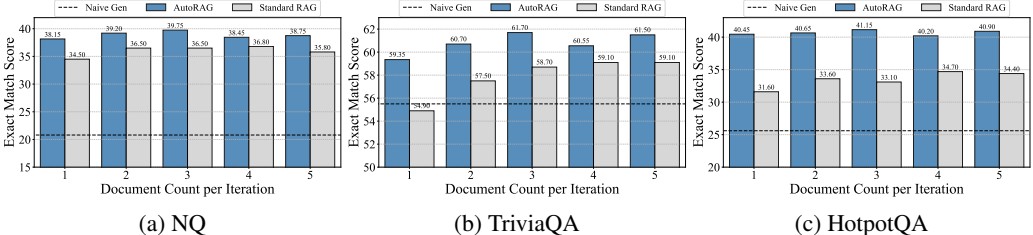

Figure 4: QA performance of Auto-RAG with varying document counts provided per iteration.

## 5 ANALYSIS

### 5.1 STRONG ADAPTABILITY TO QUESTIONS AND RETRIEVERS

In practical applications, the complexity of questions and the length of retrieved documents can vary significantly, highlighting the importance of examining Auto-RAG's adaptability to these external variations. We analyzed the proportion of iterations and performance for Auto-RAG when the retriever provides different numbers of documents at each iteration across various datasets.

First, as demonstrated in Figure 3, the proportion of terminations after a single iteration is slightly higher for NQ (Figure 3a) and TriviaQA (Figure 3b) compared to HotpotQA (Figure 3c). This difference can be attributed to the fact that NQ and TriviaQA are single-hop QA tasks, whereas HotpotQA involves multiple hops. This observation suggests that Auto-RAG is capable of adaptively adjusting the number of iterations in response to the complexity of the questions posed. Furthermore, as the quantity of documents provided in each round increases, the proportion of terminations after one iteration also rises. This indicates that Auto-RAG flexibly modulates the number of iterations based on the sufficiency of available information. Additionally, as illustrated in Figure 4, providing varying quantities of documents at each iteration has a certain impact on the overall QA performance. In these three tasks, offering three documents per iteration yields superior results, indicating that supplying Auto-RAG with appropriately sized documents is beneficial. We also compared Auto-RAG with the no-retrieval approach (Naive Gen) and Standard RAG. Auto-RAG consistently outperformed them across different document counts per iteration. Notably, Auto-RAG exhibited less performance fluctuation than Standard RAG, demonstrating its superior robustness to retrievers.

### 5.2 ABLATION STUDY

We conducted experiments to validate the effectiveness of Auto-RAG's training process, iterative reasoning, and data construction. Experimental results are shown in Table 5. First, we compared the performance of the trained Auto-RAG to a base model guided by few-shot prompts used for data synthesis (w/o training). Experimental results indicate that the trained Auto-RAG achieves superior performance, eliminating the additional inference overhead associated with the few-shot approach. To investigate the impact of iterative reasoning, we compared Auto-RAG with a base model that generated answers directly based on all documents retrieved by Auto-RAG during iterative retrieval (w/o reasoning). The experimental results are shown in Figure 6, which demonstrate that incorporating a reasoning process into Auto-RAG significantly enhances its effectiveness in solving complex problems, aligning with the conclusions of Wei et al., 2023. Furthermore, to illustrate the advantages

!

Figure 5: Experimental Results of the Ablation Study.

| Methods | NQ EM | 2Wiki F1 | TQA EM | PQA F1 | HQA F1 | WQ EM | AVG |
|---|---|---|---|---|---|---|---|
| AutoRAG | **37.9** | **48.9** | **60.9** | **47.8** | **44.9** | **25.1** | **44.3** |
| w/o training | 32.7 | 39.5 | 56.4 | 42.7 | 40.3 | 19.1 | 38.5 |
| w/o reasoning | 31.9 | 26.6 | 55.6 | 44.2 | 36.0 | 17.6 | 35.3 |
| w/o zero-shot refinement | 36.8 | 44.0 | 60.2 | 45.1 | 42.9 | 22.2 | 41.9 |

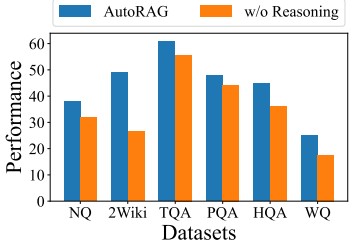

Figure 6: Ablation of reasoning.

Table 2: Performance of Auto-RAG on General Tasks.

| Methods | ARC-e Acc | ARC-c Acc | RACE-high Acc | SWAG Acc_norm | OpenBookQA Acc_norm | AVG |
|---|---|---|---|---|---|---|
| Llama-3-8B-Instruct | 93.3 | 82.0 | 81.3 | 75.3 | 43.0 | 75.0 |
| Auto-RAG | 94.2 | 84.8 | 80.3 | 75.9 | 42.8 | **75.6** |

of utilizing a zero-shot approach for query rewriting in data synthesis, we compared it with few-shot query refinement (w/o zero-shot refinement). The experimental results reveal that the zero-shot method produces more flexible and diverse queries, enhancing overall performance.

## 5.3 DATA SCALING

We investigated the performance of Auto-RAG trained on varying amounts of instructions. Specifically, we adjusted the data volume from 0.1k to 10k and evaluated the performance of the trained model on QA tasks. The experimental results are illustrated in Figure 7, indicating that approximately 0.5k of data is sufficient for the model to acquire autonomous retrieval capabilities, while increasing the data volume further enhances performance.

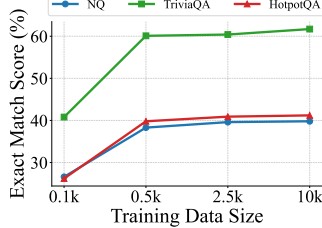

Figure 7: Performance of Auto-RAG under different amounts of training data.

## 5.4 GENERAL TASK PERFORMANCE

To evaluate the performance of Auto-RAG on general tasks, we conducted experiments on several general task evaluation benchmarks, including the AI2 Reasoning Challenge (ARC, Clark et al., 2018), ReAding Comprehension Dataset From Examinations (RACE, Lai et al., 2017), Situations With Adversarial Generations (SWAG, Zellers et al., 2018, and Open Book Question Answering (OpenBook QA Mihaylov et al., 2018). The experimental results are shown in Table 2. Auto-RAG demonstrates improved performance on ARC and SWAG, indicating that training with synthetic data can enhance LLM's reasoning abilities and capacity to tackle adversarial tasks.

## 5.5 EFFICIENCY

To demonstrate the superior performance of Auto-RAG, we compare its results with those of FLARE (Jiang et al., 2023) and Self-RAG (Asai et al., 2023), as illustrated in Figure 8. FLARE employs manually constructed rules to retrieve and revise low-probability components of the generated content. In contrast, Auto-RAG autonomously determines both when and what to retrieve, showcasing significant advantages in performance, speed, and retrieval counts. Self-RAG performs a single retrieval for short-form QA, generating one answer for each retrieved document individually while engaging in reflection, which is time-consuming and fails to consider the relevance among documents. Additionally, the number of retrievals in Self-RAG is determined by the length of the generated output. In contrast, Auto-RAG adjusts the number of iterations based on the complexity of the question and the relevance of external knowledge, resulting in superior performance and efficiency.

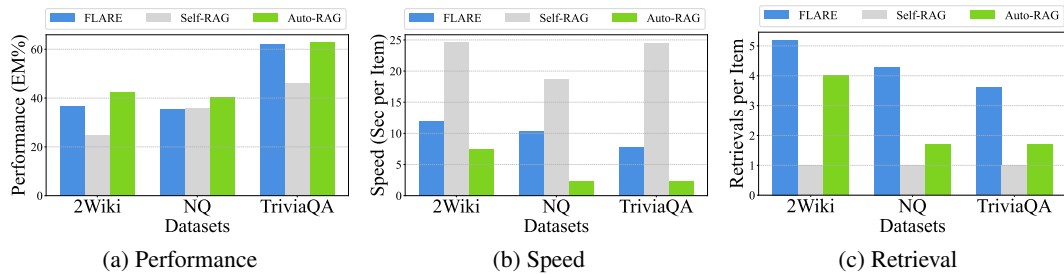

Figure 8: Comparison of Auto-RAG with FLARE and Self-RAG. Auto-RAG can autonomously adjust the number of retrievals, resulting in better performance and faster processing speeds.

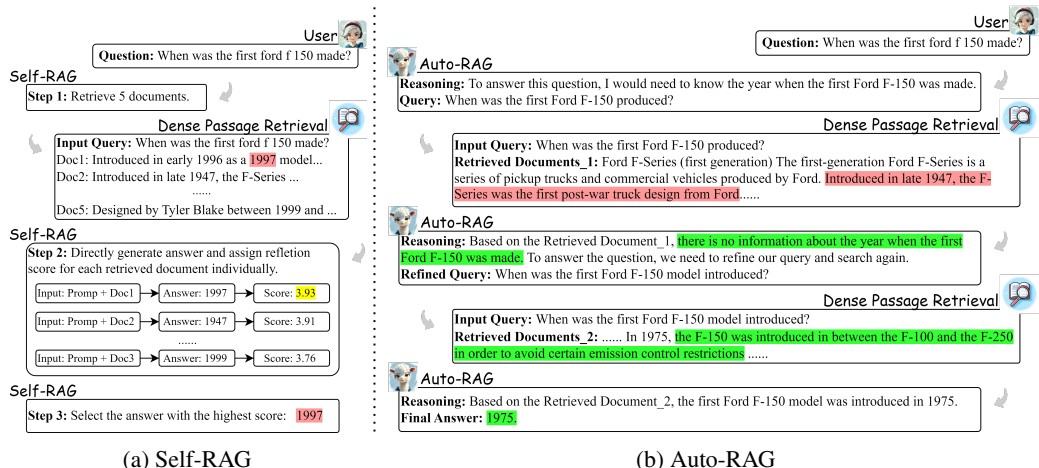

Figure 9: Case Study: Self-RAG vs. Auto-RAG. Self-RAG conducts only a single retrieval. In contrast, Auto-RAG can adaptively adjust the number of retrievals, resulting in a better performance.

## 5.6 CASE STUDY

We conducted a case study to compare Auto-RAG with Self-RAG (Asai et al., 2023), as illustrated in Figure 9. For each retrieved document, Self-RAG independently generates answers and reflects on them by predicting a reflection token, ultimately selecting the highest-scoring answer as the response. This method is not only time-consuming but also fails to account for the relevance among documents. If the existing documents are all irrelevant, Self-RAG is unable to initiate new searches to correct the erroneous answers. In contrast, Auto-RAG relies entirely on its autonomous decision-making capabilities to determine when and what to retrieve. When confronted with irrelevant documents, Auto-RAG refrains from providing an answer and continues to retrieve information until it acquires valuable knowledge, subsequently returning the answer to the user. Additionally, Auto-RAG articulates its reasoning process in natural language rather than generating reflection tokens, resulting in greater interpretability and a more intuitive user experience.

## 6 CONCLUSION

In this paper, we introduce **Auto-RAG**, an autonomous iterative retrieval model centered on the LLM's powerful decision-making capabilities. Auto-RAG interacts with the retriever through multi-turn dialogues, systematically planning retrievals and refining queries to acquire valuable knowledge until sufficient external information is obtained, at which point the results are presented to the user. To this end, we develop a method for autonomously synthesizing reasoning-based decision-making instructions in iterative retrieval and fine-tuned the latest open-source LLMs. Analysis results demonstrate that Auto-RAG not only achieves outstanding performance but also retains a high degree of interpretability, offering users a more intuitive experience.

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

# A    ADDITIONAL RESULTS

## A.1    EXPERIMENTAL RESULTS USING DIFFERENT MODELS AND VERSIONS OF WIKIPEDIA

Given that different base models and various versions of Wikipedia can impact the results, we present the outcomes from training with the Llama-3.1-8B-Instruct as the base model, as well as the results using different versions of Wikipedia as retrieval corpora.

First, we present the results from training using the Llama-3.1-8B-Instruct as the base model. The training was conducted on the same datasets used in the main experiment (generated mainly based on Llama-3-8B-Instruct). The Wikipedia 2018 dump used in the experiments followed FlashRAG(Jin et al., 2024) and DPR (Karpukhin et al., 2020). As shown in the Table 3, training with a more powerful base model yields superior results compared to those reported in the main experiment. Additionally, we utilized the Wikipedia dumps provided by Atlas (Izacard et al., 2022), which include both the 2018 and 2021 versions. We provide the results using Wikipedia 2018 dumps in Table 4 and Wikipedia 2021 dumps in Table 5.

Table 3: Experimental results for different base models.

| Methods | NQ | 2Wiki | TQA | PQA | HQA | WQ | AVG |
|---|---|---|---|---|---|---|---|
|  | EM | F1 | EM | F1 | F1 | EM | |
| *Llama-3-8B-Instruct* | | | | | | | |
| Naive Gen | 22.6 | 33.9 | 55.7 | 21.7 | 28.4 | 18.8 | 30.2 |
| Auto-RAG | **37.9** | **48.9** | **60.9** | **47.8** | **44.9** | **25.1** | **44.3** |
| *Llama-3.1-8B-Instruct* | | | | | | | |
| Naive Gen | 23.9 | 30.3 | 56.9 | 28.6 | 29.0 | 16.9 | 30.9 |
| Auto-RAG | **40.5** | **51.4** | **62.7** | **49.3** | **48.5** | **23.4** | **46.0** |

Table 4: Experimental results using Wikipedia Dump 2018 provided by Atlas (Izacard et al., 2022).

| Methods | NQ | 2Wiki | TQA | PQA | HQA | WQ | AVG |
|---|---|---|---|---|---|---|---|
|  | EM | F1 | EM | F1 | F1 | EM | |
| *Llama-3-8B-Instruct* | | | | | | | |
| Naive Gen | 22.6 | 33.9 | 55.7 | 21.7 | 28.4 | 18.8 | 30.2 |
| Auto-RAG | **38.9** | **59.9** | **60.6** | **52.7** | **47.0** | **25.1** | **47.4** |
| *Llama-3.1-8B-Instruct* | | | | | | | |
| Naive Gen | 23.9 | 30.3 | 56.9 | 28.6 | 29.0 | 16.9 | 30.9 |
| Auto-RAG | **42.0** | **62.1** | **62.0** | **54.7** | **51.7** | **21.9** | **49.1** |

Table 5: Experimental results using Wikipedia Dump 2021 provided by Atlas (Izacard et al., 2022).

| Methods | NQ | 2Wiki | TQA | PQA | HQA | WQ | AVG |
|---|---|---|---|---|---|---|---|
|  | EM | F1 | EM | F1 | F1 | EM | |
| *Llama-3-8B-Instruct* | | | | | | | |
| Naive Gen | 22.6 | 33.9 | 55.7 | 21.7 | 28.4 | 18.8 | 30.2 |
| AutoRAG | **35.2** | **59.2** | **60.5** | **51.5** | **44.7** | **25.1** | **46.0** |
| *Llama-3.1-8B-Instruct* | | | | | | | |
| Naive Gen | 23.9 | 30.3 | 56.9 | 28.6 | 29.0 | 16.9 | 30.9 |
| AutoRAG | **38.9** | **62.3** | **62.5** | **53.6** | **49.3** | **21.0** | **47.9** |

Table 6: Experimental results of closed-source models.

| Method | Model | NQ | 2Wiki | TQA | PQA | HQA | WQ | AVG |
| | | EM | F1 | EM | F1 | F1 | EM | |
| --- | --- | --- | --- | --- | --- | --- | --- | --- |
| | | *No Retrieval* | | | | | | |
| Naive-Gen | gpt-4o-2024-08-06 | 16.9 | 43.2 | **69.1** | 48.2 | **48.3** | 15.1 | 40.1 |
| | gpt-4o-mini-2024-07-18 | 19.2 | 31.7 | 59.6 | 35.1 | 37.9 | 19.9 | 33.9 |
| | Llama-3-8B-Instruct | 20.9 | 25.7 | 54.0 | 26.3 | 27.1 | 20.1 | 29.0 |
| | | *Standard Retrieval* | | | | | | |
| Standard RAG | gpt-4o-2024-08-06 | 14.0 | 36.2 | 58.7 | 45.6 | 46.8 | 13.9 | 35.9 |
| | gpt-4o-mini-2024-07-18 | 29.9 | 34 | 61.3 | **49.6** | 45.6 | 19.5 | 40.0 |
| | Llama-3-8B-Instruct | **35.1** | 19.1 | 56.9 | 47.5 | 35.6 | 16.3 | 35.1 |
| | | *Autonomous Retrieval* | | | | | | |
| Auto-RAG | Llama-3-8B-Instruct | 34.2 | **47.9** | 58.6 | 48.4 | 45.7 | **23.4** | **43.0** |

Table 7: Experimental results with different knowledge provision orders. "Parametric-External" refers to providing external knowledge first, followed by parametric knowledge, while "External-Parametric" denotes the reverse order.

| Order | NQ | 2Wiki | TQA | HQA | PQA | WQ | AVG |
| | EM | F1 | EM | F1 | F1 | EM | |
| --- | --- | --- | --- | --- | --- | --- | --- |
| no-parametric | 37.7 | 39.8 | 60.1 | 42.0 | **46.9** | 22.6 | 41.5 |
| parametric-external | 26.7 | 37.4 | 54.3 | 33.8 | 34.6 | 18.2 | 34.2 |
| external-parametric | **37.9** | **48.9** | **60.9** | **47.8** | 44.9 | **25.1** | **44.3** |

## A.2 Comparison with Closed-source Models

To further demonstrate the effectiveness of Auto-RAG, we present results comparing it with closed-source models, such as GPT-4o. Due to budget and time constraints, we sampled 1,000 samples from each dataset and compared the performance of our method with that of closed-source models. The random seed was set to 0. The experimental results are shown in Table 6. Firstly, the average performance of Auto-RAG is the best. Secondly, GPT-4o demonstrated better performance without retrieval, while GPT-4o-mini showed improved performance after retrieval. It indicates that for a well-trained model, the quality of its parametric knowledge may be higher than that of external knowledge. Therefore, providing external knowledge may degrade its performance. Enhancing the model's ability to resist irrelevant information is crucial. Auto-RAG autonomously adjusts its retrieval strategy based on the availability of external knowledge. When external knowledge is useful, it answers sub-questions, generates new queries, or derives a conclusion. If the external knowledge is not useful, it refuses to answer and re-initiates the search process.

## A.3 Impact of the Order of External and Parametric Knowledge

As mentioned in Section 3.3, during the first $T$ iterations, external knowledge is provided to the model; in the subsequent $T^{PK}$ iterations, parametric knowledge is provided. We will first explain the rationale behind this design and then present experiments to validate it.

The reason we first provide external knowledge to the model and then parameterized knowledge is as follows:

- As shown in the main experiment in Table 1, the model performs better on average when external knowledge is provided (Standard RAG vs Naive Gen). This suggests that, for LLaMA-3-8B-Instruct, external knowledge may be more valuable.
- The knowledge generated by LLM is **highly misleading** (Xie et al., 2024). LLMs are capable of generating more coherent yet fabricated knowledge that is convincing to LLMs.

Table 8: Distributions of iteration counts when the external and parametric knowledge are provided in different orders.

| Order | Distributions of Iteration Counts | | | | | |
|---|---|---|---|---|---|---|
| | **1** | **2** | **3** | **4** | **5** | **6** |
| **no-parametric** | 44.65% | 47.56% | 2.94% | 0.97% | 0.55% | 0.14% |
| **parametric-external** | 82.08% | 8.98% | 0.50% | 0.30% | 0.28% | 6.70% |
| **external-parametric** | 44.65% | 47.56% | 2.94% | 0.97% | 0.58% | 2.33% |

Table 9: Comparison between Auto-RAG and Self-RAG. Accuracy is the reported metric

| Method | TriviaQA | PopQA |
|---|---|---|
| Self-RAG | 69.3 | 55.8 |
| Auto-RAG | **70.2** | **59.7** |

Next, we designed experiments to examine the impact of providing parametric knowledge and the order in which parametric and external knowledge are presented. Experimental results are shown in Table 7. To evaluate the effect of providing parametric knowledge on Auto-RAG performance (no-parametric), we kept the maximum number of iterations the same and provided only external knowledge. The results (no-parametric vs. external-parametric) show that using only external knowledge yields good performance, and supplementing with parametric knowledge further enhances the results. To assess the impact of the order in which the two types of knowledge are provided, we swapped the sequence of knowledge presentation while keeping all other settings the same. The results (parametric-external vs. external-parametric) indicate that providing external knowledge first, followed by parametric knowledge, leads to better performance.

To demonstrate that the model-generated parametric knowledge is more relevant and convincing, we analyzed the distribution of iteration counts when different types of knowledge are provided in varying sequences on NQ. The experimental results are shown in Table 8. When parameter knowledge is provided first, Auto-RAG requires fewer iterations. However, the QA performance is suboptimal in this case, suggesting that the LLM may generate plausible yet fabricated knowledge. This conclusion is consistent with the findings of Xie et al. (2024).

## A.4 ADDITIONAL COMPARISON WITH SELF-RAG

Since the evaluation scope and metrics used in the Self-RAG paper differ from those of our main experiments, we conducted experiments following their original setup. Specifically, we use the long-tail subset, consisting of 1,399 rare entity queries whose monthly Wikipedia page views are less than 100 from PopQA. We evaluated the performance using Accuracy (i.e., whether the standard answer appeared in the Final Answer). The results, as shown in Table 9, demonstrate that Auto-RAG consistently outperforms Self-RAG.

## B HYPERPARAMETER SETTINGS

Table 10: Hyperparameters used in main experiments and analysis. $T$ represents the maximum number of interactions with the retriever. $T^{PK}$ denotes the max number of times parametric knowledge is requested. "Docs num per iter" refers to the number of documents provided in each iteration.

| Hyperparameters | NQ | 2Wiki | TriviaQA | PopQA | HotpotQA | WebQA |
|---|---|---|---|---|---|---|
| $T$ | 5 | 10 | 5 | 5 | 5 | 5 |
| $T^{PK}$ | 5 | 5 | 5 | 5 | 5 | 5 |
| Doc num per iter | 3 | 2 | 3 | 2 | 3 | 1 |

Since Auto-RAG can autonomously determine the number of iterations in most cases, we do not need to explore all possible maximum iterations exhaustively. Instead, we set a relatively flexible maximum iteration limit to ensure timely termination when the retriever fails to provide useful knowledge. Additionally, a key hyperparameter for the retriever is the number of documents provided per iteration. Providing more documents per round increases the recall of useful knowledge but also raises the difficulty for the model in extracting relevant information. We tuned the number of documents provided per iteration by sampling 2,000 examples from the validation set. The settings for the above hyperparameters are shown in Table 10, and the same hyperparameters are used for all analysis experiments.

## C PROMPT TEMPLATES AND EXAMPLES

### C.1 PROMPT FOR ELICITING REASONING

We construct few-shot prompts for eliciting reasoning process (Asai et al., 2023; Jiang et al., 2023). As the synthetic data is generated based on 2WikiMultihop and NQ, we developed two distinct prompts accordingly. For 2WikiMultihopQA, we modified the 8-shot examples used by FLARE to create the demonstrations. Due to the length of the examples, we present two from the 8-shot demonstrations and omit the retrieved documents:

```
Answer the following questions by retrieving external knowledge. Extract
    useful information from each retrieved document. If the information
    is insufficient or irrelevant, refine your query and search again
    until you are able to answer the question:

###

Question: Do director of film Coolie No. 1 (1995 Film) and director of
    film The Sensational Trial have the same nationality?

Analysis: To answer this question, I would need to know the nationality
    of the director of the film "Coolie No. 1" (1995) and the nationality
     of the director of the film "The Sensational Trial".

Initial Query: Coolie No. 1 (1995 Film)

Retrieved Document_1: <retrieved_document_1>

Intermediate Answer_1: Based on the Retrieved Document_1, the director of
     film Coolie No. 1 (1995 Film) is David Dhawan. However, there is no
    information about the nationality of David Dhawan in this document.
    To answer the question, we need to find the nationality of David
    Dhawan.

Refined Query: David Dhawan

Retrieved Document_2: <retrieved_document_2>

Intermediate Answer_2: Based on the Retrieved Document_2, David Dhawan is
     an Indian film director. There is no information about the
    nationality of the director of film The Sensational Trial in this
    document. To answer the question, we need to find the nationality of
    the director of film The Sensational Trial.

Refined Query: The Sensational Trial

Retrieved Document_3: <retrieved_document_3>

Intermediate Answer_3: Based on the Retrieved Document_3, there is no
    information about the director of film The Sensational Trial. The
    document is about an American doo-wop group, not a film. We need to
    refine our query and search again.
```

```
1080    Refined Query: The Sensational Trial (film)
1081
1082    Retrieved Document_4: <retrieved_document_4>
1083
1084    Intermediate Answer_4: Based on the Retrieved Document_4, there is no
1085        information about the director of film The Sensational Trial. The
            document is about the members of a doo-wop group, not a film. We need
1086        to refine our query and search again.
1087
1088    Refined Query: The Sensational Trial (film) director
1089
1090    Retrieved Document_5: <retrieved_document_5>
1091
1092    Intermediate Answer_5: Based on the Retrieved Document_5, the director of
            film The Sensational Trial is Karl Freund. To answer the question,
1093        we need to find Karl Freund's nationality.
1094
1095    Refined Query: Karl Freund
1096
1097    Retrieved Document_6: <retrieved_document_6>
1098
1099    Intermediate Answer_6: Based on the Retrieved Document_6, Karl Freund was
            a German Jewish cinematographer and film director. Comparing the
1100        nationalities of David Dhawan and Karl Freund, David Dhawan is Indian
1101        , while Karl Freund was German Jewish. Therefore, the director of
1102        film Coolie No. 1 (1995 Film) and the director of film The
            Sensational Trial do not have the same nationality.
1103
1104    Final Answer: No.
1105
        ###
1106
1107    Question: When did the director of film Hypocrite (Film) die?
1108
1109    Analysis: To answer this question, I would need to know the name of the
1110        director of the film "Hypocrite" (Film).
1111
        Initial Query: Director of film Hypocrite (Film)
1112
1113    Retrieved Document_1: <retrieved_document_1>
1114
1115    Intermediate Answer_1: Based on the Retrieved Document_1, the director of
            film Hypocrite (Film) is Miguel Morayta. However, there is no
1116        information about the date of death of Miguel Morayta in this
1117        document. To answer the question, we need to find the date of death
1118        of Miguel Morayta.
1119
1120    Refined Query: Miguel Morayta
1121
1122    Retrieved Document_2: <retrieved_document_2>
1123    Intermediate Answer_2: Based on the Retrieved Document_2, Miguel Morayta
1124        died on 19 June 2013. Therefore, the director of film Hypocrite (Film
            ) died on 19 June 2013.
1125
1126    Final Answer: 19 June 2013.
1127
1128    ###
1129
```

and the prompt for NQ is shown below:

```
Answer the following questions by retrieving external knowledge. Extract
    useful information from each retrieved document. If the information
    is insufficient or irrelevant, refine your query and search again
    until you are able to answer the question:
```

```
###

Question: Who does the voice of susan in monsters vs aliens?

Analysis: To answer this question, I would need to know the voice actor
    for the character Susan in the movie Monsters vs. Aliens.

Initial Query: Monsters vs. Aliens

Retrieved Document_1: <retrived_document_1>

Intermediate Answer_1: Based on the Retrieved Document_1, the voice of
    Susan in Monsters vs. Aliens is Reese Witherspoon.

Final Answer: Reese Witherspoon.

###

Question: Who played jason in mighty morphin power rangers?

Analysis: To answer this question, I would need to know the actor who
    played Jason in Mighty Morphin Power Rangers.

Initial Query: Mighty Morphin Power Rangers

Retrieved Document_1: <retrieved_document_1>

Intermediate Answer_1: Based on the Retrieved Document_1, there is no
    information about the actor who played Jason in Mighty Morphin Power
    Rangers. To answer the question, we need to refine our query and
    search again.

Refined Query: Mighty Morphin Power Rangers Jason

Retrieved Document_2: <retrieved_document_2>

Intermediate Answer_2: Based on the Retrieved Document_2, the actor who
    played Jason in Mighty Morphin Power Rangers is Austin St. John.

Final Answer: Austin St. John.

###
```

## C.2 PROMPT TEMPLATE FOR NAIVE GENERATION AND STANDARD RAG

Following Jin et al. (2024), we utilize the prompt template for Naive Generation as follows:

```
<|begin_of_text|><|start_header_id|>system<|end_header_id|>

Answer the question based on your own knowledge. Only give me the answer
    and do not output any other words.<|eot_id|><|start_header_id|>user<|
    end_header_id|>

Question: {question}<|eot_id|><|start_header_id|>assistant<|end_header_id
    |>
```

and the prompt template used for Standard RAG is shown below:

```
<|begin_of_text|><|start_header_id|>system<|end_header_id|>

Answer the question based on the given document.Only give me the answer
    and do not output any other words.
The following are given documents.
```

```
Doc {doc_id}(Title: {doc_title}) {doc_text}
Doc {doc_id}(Title: {doc_title}) {doc_text}
Doc {doc_id}(Title: {doc_title}) {doc_text}
Doc {doc_id}(Title: {doc_title}) {doc_text}
Doc {doc_id}(Title: {doc_title}) {doc_text}

<|eot_id|><|start_header_id|>user<|end_header_id|>

Question: {question}<|eot_id|><|start_header_id|>assistant<|end_header_id
    |>
```

### C.3 PROMPT TEMPLATE FOR FEW–SHOT QUERY REWRITING

The prompt for few-shot query rewriting on NQ used in Section 5.2 is shown below:

```
Question: How many episodes in a season of vampire diaries?
Model Analysis: To answer this question, I would need to know the number
    of episodes in a season of The Vampire Diaries.
Query: The Vampire Diaries
###
Question: Who played jason in mighty morphin power rangers?
Model Analysis: Based on the Retrieved Document_1, there is no
    information about the actor who played Jason in Mighty Morphin Power
    Rangers. To answer the question, we need to refine our query and
    search again.
Query: Mighty Morphin Power Rangers Jason
###
Question: Who developed an explanation for the photoelectric effect?
Model Analysis: Based on the Retrieved Document_2, there is no
    information about who developed an explanation for the photoelectric
    effect. To answer the question, we need to refine our query and
    search again.
Query: Photoelectric Effect Explanation
###
Question: District of maharashtra that are part of red corridor?
Model Analysis: Based on the Retrieved Document_2, there is no
    information about the districts of Maharashtra that are part of the
    Red Corridor. To answer the question, we need to refine our query and
     search again.
Query: Red Corridor in Maharashtra districts
###
```

and that for 2WikiMultihopQA is shown below:

```
Question: When did the director of film Hypocrite (Film) die?
Model Analysis: To answer this question, I would need to know the name of
     the director of the film "Hypocrite" (Film).
Query: Director of film Hypocrite (Film)
###
Question: Are both Kurram Garhi and Trojkrsti located in the same country
    ?
Model Analysis: Based on the Retrieved Document_1, Kurram Garhi is
    located in Pakistan. However, there is no information about Trojkrsti
     in this document. To answer the question, we need to find the
    location of Trojkrsti.
Query: Trojkrsti
###
Question: Which film has the director died later, The Gal Who Took the
    West or Twenty Plus Two?
Model Analysis: Based on the Retrieved Document_3, Frederick de Cordova
    died on September 15, 2001. There is no information about the date of
     death of the director of film Twenty Plus Two in this document. To
    answer the question, we need to find the date of death of the
    director of film Twenty Plus Two.
```

```
Query: Twenty Plus Two
###
Question: Do director of film Coolie No. 1 (1995 Film) and director of
    film The Sensational Trial have the same nationality?
Model Analysis: Based on the Retrieved Document_2, David Dhawan is an
    Indian film director. There is no information about the nationality
    of the director of film The Sensational Trial in this document. To
    answer the question, we need to find the nationality of the director
    of film The Sensational Trial.
Query: The Sensational Trial
###
```

### C.4 Prompt Template for Eliciting Parametric Knowledge

The prompt template used for eliciting parametric knowledge is shown below:

```
Your task is to generate one corresponding wikipedia document based on
    the given query to help the LLM answer questions.

Demostrations:

Origin Question: How many episodes in a season of vampire diaries?

Query: The Vampire Diaries episode count

Document: The Vampire Diaries has a total of 171 episodes over 8 seasons.
     The show's first season had 22 episodes, the second season had 22
    episodes, the third season had 22 episodes, the fourth season had 23
    episodes, the fifth season had 22 episodes, the sixth season had 22
    episodes, the seventh season had 22 episodes, and the eighth season
    had 16 episodes.

###

Origin Question: Who developed an explanation for the photoelectric
    effect?

Query: Photoelectric Effect Explanation

Document: To make sense of the fact that light can eject electrons even
    if its intensity is low, Albert Einstein proposed that a beam of
    light is not a wave propagating through space, but rather a
    collection of discrete wave packets (photons), each with energy hv.
    This shed light on Max Planck's previous discovery of the Planck
    relation (E = hv) linking energy (E) and frequency (v) as arising
    from quantization of energy. The factor h is known as the Planck
    constant. In 1887, Heinrich Hertz discovered that electrodes
    illuminated with ultraviolet light create electric sparks more easily
    . In 1900, while studying black-body radiation, the German physicist
    Max Planck suggested that the energy carried by electromagnetic waves
     could only be released

###

Origin Question: District of maharashtra that are part of red corridor?

Query: Red Corridor in Maharashtra districts

Document: The Red Corridor in Maharashtra includes the following
    districts: Chandrapur, Gondia, and Gadchiroli.

###

Origin Question: Who played jason in mighty morphin power rangers?
```

```
Query: Mighty Morphin Power Rangers Jason

Document: from Dairanger were featured in the second season while only
    the Kakuranger mecha was featured in the third season, though the
    Kakuranger costumes were later used for the mini-series Mighty
    Morphin Alien Rangers. The series was produced by MMPR Productions
    and distributed by Saban Entertainment, while the show's merchandise
    was produced and distributed by Bandai Entertainment. The series was
    well known for its campy tone. In 2010, a re-version of Mighty
    Morphin Power Rangers, with a revised new look of the original 1993
    logo, comic book-referenced graphics, and extra alternative visual
    effects, was broadcast on ABC Kids, and Bandai produced brand new
    toys to coincide with the series. Only the first 32 of season one's
    60 episodes were remade.

###

Origin Question: {}

Query: {}

Document:
```

## C.5 PROMPT TEMPLATE FOR ZERO-SHOT QUERY REWRITING

The prompt used for zero-shot query rewriting introduced in Section 3.1.1 is shown below:

```
Please generate a query based on the problem and the model analysis. The
    retrieval engine is Dense Passage Retrieval. Please generate a
    suitable query for this retriever. Please follow the format: Query:
    your query for Dense Passage Retrieval.

Question: {}

Model Analysis: {}

Query:
```

## C.6 EXAMPLES OF SYNTHESIZED INSTRUCTIONS

In this section, we provide several examples of constructed training data.

```
Example 1:

Question: What is the date of birth of the director of film Point Of
    Betrayal?
Analysis: To answer this question, I would need to know the director of
    the film "Point Of Betrayal" and his date of birth.\n\nInitial Query:
     Date of birth of the director of the film "Point Of Betrayal"
Retrieved Document_1: Point of Betrayal Point of Betrayal is a 1995 film
    about a man (Rick Johnson) trying to drive his mother (Dina Merrill)
    insane in order to get her money. The film was directed by Richard
    Martini and produced by Jonathan D. Krane. A man (Rick Johnson) tries
     to drive his mother (Dina Merrill) insane in order to get her money.
     He hires a nurse (Rebecca Broussard) for his mother but faces
    opposition from his step father (Rod Taylor). At the request of the
    director, Rod Taylor assisted writing some scenes and helped
    choreograph a fight between him and Rick Johnson.
Intermediate Answer_1: Based on the Retrieved Document_1, the director of
     film Point Of Betrayal is Richard Martini. However, there is no
    information about the date of birth of Richard Martini in this
    document. To answer the question, we need to find the date of birth
```

of Richard Martini.\n\nRefined Query: What is the date of birth of
   Richard Martini, the director of Point Of Betrayal?
Retrieved Document_2: Point of Betrayal Point of Betrayal is a 1995 film
   about a man (Rick Johnson) trying to drive his mother (Dina Merrill)
   insane in order to get her money. The film was directed by Richard
   Martini and produced by Jonathan D. Krane. A man (Rick Johnson) tries
    to drive his mother (Dina Merrill) insane in order to get her money.
    He hires a nurse (Rebecca Broussard) for his mother but faces
   opposition from his step father (Rod Taylor). At the request of the
   director, Rod Taylor assisted writing some scenes and helped
   choreograph a fight between him and Rick Johnson.
Intermediate Answer_2: Based on the Retrieved Document_2, there is no
   information about the date of birth of Richard Martini. We need to
   refine our query and search again.\n\nRefined Query: What is the date
    of birth of Richard Martini, the director of the film Point Of
   Betrayal?
Retrieved Document_3: people no longer on the planet and compares these
   accounts to the near death experiences and afterlife reports from
   subjects under hypnosis. Martini has taught film directing at Loyola
   Marymount University, the Maine Media Workshops, and the John Felice
   Rome Center. He is married and has two children. The family lives in
   Santa Monica, California. Richard Martini Richard Martini (born 12
   March 1955) is an American film director, producer, screenwriter and
   freelance journalist. Martini was born in 1955 and grew up in
   Northbrook, Illinois. He attended local public schools. He graduated
   "magna cum laude" from Boston University with a degree
Intermediate Answer_3: Based on the Retrieved Document_3, Richard Martini
    was born on 12 March 1955. Therefore, the date of birth of the
   director of film Point Of Betrayal is 12 March 1955.\n\nFinal Answer:
    12 March 1955.

Example 2:

Question: What is the award that the director of film Against All Odds
   (1984 Film) won?
Analysis: To answer this question, I would need to know the director of
   the film "Against All Odds" (1984) and the award he won.\n\nInitial
   Query: Award won by the director of the film "Against All Odds"
   (1984)
Retrieved Document_1: Against All Odds (1984 film) Against All Odds is a
   1984 American romantic neo-noir thriller film. A remake of "Out of
   the Past" (1947), the film that currently holds a "Fresh" score of 67
   *% on review aggregation website Rotten Tomatoes. "Against All Odds"*
   *was directed by Taylor Hackford and stars Rachel Ward, Jeff Bridges*
   *and James Woods alongside Jane Greer (who had starred in "Out of the*
   *Past"), Alex Karras, Richard Widmark and Dorian Harewood. The film*
   *revolves around an aging American football star who is hired by a*
   *mobster to find his girlfriend. The film\'s soundtrack, nominated for*
    *a*
Intermediate Answer_1: Based on the Retrieved Document_1, the director of
    film Against All Odds (1984) is Taylor Hackford. However, there is
   no information about the award won by Taylor Hackford in this
   document. To answer the question, we need to find the award won by
   Taylor Hackford.\n\nRefined Query: What award did Taylor Hackford win
   ?
Retrieved Document_2: Taylor Hackford Taylor Edwin Hackford (born
   December 31, 1945) is an American film director and former president
   of the Directors Guild of America. He won the Academy Award for Best
   Live Action Short Film for "Teenage Father" (1979). Hackford went on
   to direct a number of highly regarded feature films, most notably "An
    Officer and a Gentleman" (1982) and "Ray" (2004), the latter of
   which saw him nominated for the Academy Award for Best Director and
   Academy Award for Best Picture. Hackford was born in Santa Barbara,
   California, the son of Mary (nee Taylor), a waitress, and Joseph
   Hackford. He

```
Intermediate Answer_2: Based on the Retrieved Document_2, Taylor Hackford
    won the Academy Award for Best Live Action Short Film for "Teenage
    Father" (1979).\n\nFinal Answer: Academy Award for Best Live Action
    Short Film.
```

## C.7 EXAMPLES OF AUTO-RAG'S OUTPUT

In this section, we present sample outputs from Auto-RAG.

Table 11: Output Example 1

| | |
|---|---|
| **Input** Question: When did John V, Prince Of Anhalt-Zerbst's father die? | |
| **Output** Analysis: To answer this question, I would need to know the name of John V, Prince Of Anhalt-Zerbst's father. 
 Initial Query: Who is John V, Prince Of Anhalt-Zerbst's father? | |
| **Input** Retrieved Document_1: John V, Prince of Anhalt-Zerbst John V of Anhalt-Zerbst (Dessau, 4 September 1504 - Zerbst, 4 February 1551), was a German prince of the House of Ascania and ruler of the principality of Anhalt-Dessau. From 1544, he assumed rule of the re-created principality of Anhalt-Zerbst. John was the second (but eldest surviving) son of Ernest I, Prince of Anhalt-Dessau, by his wife Margarete, daughter of Henry I, Duke of Munsterberg-Oels, and granddaughter of George of Podebrady, King of Bohemia. Upon the death of his father in 1516, John and his brothers George III and Joachim I inherited Anhalt-Dessau as co-rulers John VI, Prince of Anhalt-Zerbst John VI of Anhalt-Zerbst (Zerbst, 24 March 1621 - Zerbst, 4 July 1667), was a German prince of the House of Ascania and ruler of the principality of Anhalt-Zerbst. He was the only son of Rudolph, Prince of Anhalt-Zerbst, by his second wife Magdalene, daughter of John VII, Count of Oldenburg. John succeeded his father in Anhalt-Zerbst at only four months of age; during his long minority, his paternal uncle Augustus of Anhalt-Plotzkau acted as regent in the principality. John's education was supervised primarily by his mother. Political instability caused by warfare during the Thirty union with the widowed daughter of the Elector of Brandenburg was a high honor for John, and he decided to celebrate the wedding with great pomp. But by that time, he was in poor health; finally, in 1544, he suffered a stroke. His relations with Margarete worsened during the following years; in 1550 John ordered the temporary arrest of his wife, but she fled. John and Margarete had six children: John V, Prince of Anhalt-Zerbst John V of Anhalt-Zerbst (Dessau, 4 September 1504 - Zerbst, 4 February 1551), was a German prince of the House of Ascania and ruler of | |
| **Output** Intermediate Answer_1: Based on the Retrieved Document_1, John V, Prince Of Anhalt-Zerbst's father is Ernest I, Prince of Anhalt-Dessau. To answer the question, we need to find the date of death of Ernest I, Prince of Anhalt-Dessau. 
 Refined Query: When did Ernest I, Prince of Anhalt-Dessau die? | |

| | |
|---|---|
| **Input** Retrieved Document_2: Ernest I, Prince of Anhalt-Dessau Ernest I, Prince of Anhalt-Dessau (died Dessau, 12 June 1516), was a German prince of the House of Ascania and ruler of the principality of Anhalt-Dessau. He was the second son of George I, Prince of Anhalt-Dessau, yet the first born by his fourth wife Anna, daughter of Albert VIII, Count of Lindow-Ruppin. In 1473, after the death of his father, Ernest inherited the principality of Anhalt-Dessau alongside his younger brothers George II, Sigismund III, and Rudolph IV. Following the family law of the House of Ascania, the accession took place without any division of territories. The deaths of Sigismund III in 1487, George II in 1509, and Rudolph IV in 1510 without surviving male issue left Ernest as the sole ruler of Anhalt-Dessau until his death. In Cottbus on 20 January 1494 Ernest married Margarete (b. Breslau, 25 August 1473 - d. Dessau, 28 June 1530), daughter of Henry I, Duke of Munsterberg-Oels, and granddaughter of George of Podebrady, King of Bohemia. They had four sons: Ernest I, Prince of Anhalt-Dessau Ernest I, Prince of Anhalt-Dessau (died Dessau, 12 June 1516), was a German prince of the House of Ascania and ruler of the Prince Joachim Ernest died in Dessau on 6 December 1586. Since succession in the territories of Anhalt was not governed by the rules of primogeniture, Joachim Ernest's seven sons shared rule of the territories of Anhalt until 1603, when the five surviving sons divided their lands among themselves. Joachim Ernest, Prince of Anhalt Joachim Ernest of Anhalt (21 October 1536 - 6 December 1586), was a German prince of the House of Ascania, ruler of the principality of Anhalt-Zerbst from 1551, and from 1570 sole ruler of all the Anhalt lands. Joachim Ernest was born in Dessau on 21 October |
| **Output** Intermediate Answer_2: Based on the Retrieved Document_2, Ernest I, Prince of Anhalt-Dessau died on 12 June 1516. Final Answer: 12 June 1516. |

Table 12: Output Example 2

| |
|---|
| **Input** Question: Where was the place of death of Anastasia Of Serbia's husband? |
| **Output** Analysis: To answer this question, I would need to know the name of Anastasia Of Serbia's husband and his place of death. Initial Query: Who is Anastasia Of Serbia's husband and where did he die? |
| **Input** Retrieved Document_1: Anastasia of Serbia Ana (); 1196-d. June 22, 1200) was the princess consort of the Serbian Principality as the wife of Stefan Nemanja (r. 1166-1196). She was of noble descent. Ana took monastic vows in 1196 and was baptized "Anastasia", after Anastasia of Sirmium. She is venerated by the Serbian Orthodox Church as Saint Anastasij with her feast day on June 22 (Julian calendar). Her origins have never been concluded. The oldest source mentioning her origin was Domentijan (c. 1210 - after 1264), who said of her: "a great princess, daughter of the Byzantine Emperor Romanos", only Romanos IV Diogenes ruled 1068-71, making this genealogy impossible. Through the marriage with Stefan Nemanja, they had three sons and three daughters: Anastasia of Serbia Ana (); 1196-d. June 22, 1200) was the princess consort of the Serbian Principality as the wife of Stefan Nemanja (r. 1166-1196). She was of noble descent. Ana took monastic vows in 1196 and was baptized "Anastasia", after Anastasia of Sirmium. She is venerated by the Serbian Orthodox Church as Saint Anastasija with her feast day on June 22 (Julian calendar). Her origins have never been concluded. The oldest source mentioning her origin was Anastasia of Sirmium Saint Anastasia (died 25 December, A.D. 304) is a Christian saint and martyr who died at Sirmium in the Roman province of Pannonia Secunda (modern Serbia). In the Orthodox Church, she is venerated as St. Anastasia the "Pharmakolytria", i.e. "Deliverer from Potions" (). Concerning Anastasia little is reliably known, save that she died in the persecutions of Diocletian; most stories about her date from several centuries after her death and make her variously a Roman or Sirmian native and a Roman citizen of patrician rank. One legend makes her the daughter of a certain Praetextatus and the |
| **Output** Intermediate Answer_1: Based on the Retrieved Document_1, Anastasia Of Serbia's husband is Stefan Nemanja. However, there is no information about the place of death of Stefan Nemanja in this document. To answer the question, we need to find the place of death of Stefan Nemanja. Refined Query: Where did Stefan Nemanja die? |

**Input** Retrieved Document_2: Stefan Nemanja Stefan Nemanja ( ; 1113 - 13 February 1199) was the Grand Prince ("Veliki zupan") of the Serbian Grand Principality (also known as "Rascia") from 1166 to 1196. A member of the Vukanovic dynasty, Nemanja founded the Nemanjic dynasty, and is remembered for his contributions to Serbian culture and history, founding what would evolve into the Serbian Empire, as well as the national church. According to the Serbian Academy of Sciences and Arts, Nemanja is also among the most remarkable Serbs for his literary contributions and altruistic attributes. In 1196, after three decades of warfare Nemanja Nemanja () is a masculine Serbian given name. It is derived from the by-name borne by the founder of the Nemanjic dynasty, Stefan Nemanja (1114-1199), a Serbian grand prince who was venerated as a saint after his death. Etymologically, many think the name most likely derives from a meaning "without possessions", from Serbian "nemati" "to have not", but that is not true. The name origins from "ne manuti" "not to let go" in terms of being persistent and stubborn. Another theory is that the name Nemanja means dragon/demon person. This stems from the two root parts of the name. Byzantine representatives on 14 February 1190 in Adrianople. In 1190, the new Byzantine Emperor Isaac II Angelos prepared a massive and experienced army to strike against Nemanja. The same year, Nemanja finished his magnificent Virgin's Church in the Studenica Monastery out of white marble which later became the Nemanjic dynasty's hallmark. Also in 1190 his brother Miroslav died of old age, so Stefan Nemanja temporarily assigned his pious youngest son Rastko as the new Prince of Zahumlje in Ston. In fall of 1191, the well-prepared Byzantine army, led by the Emperor himself, crushed Nemanja's forces in South Morava. Stefan Nemanja

**Output** Intermediate Answer_2: Based on the Retrieved Document_2, Stefan Nemanja died on 13 February 1199. There is no information about the place of death in this document. To answer the question, we need to refine our query and search again.
Refined Query: Where did Stefan Nemanja die?

**Input** Retrieved Document_3: ruined Eastern Orthodox Monastery of Hilandar which the Byzantine Emperor had given to the Serbian people and which became the heart of Serbian spirituality. Knowing his death was near in his 86th year, Simeon asked to be placed on a mat in front of the icon of the Virgin Hodegetria with a stone for his pillow. He died in front of his son Sava and other monks, on 13 February 1200. He was buried in the grounds of Hilandar monastery. His last words requested that Sava take his remains to Serbia, "when God permits it, after a certain period of married to a Serb noblewoman by the name of Ana. They had three sons and three daughters: And possibly: Stefan Nemanja founded, restored and reconstructed several monasteries. He also established the Rascian architectural style, that spanned from 1170-1300. Stefan Nemanja Stefan Nemanja (; 1113 - 13 February 1199) was the Grand Prince ("Veliki zupan") of the Serbian Grand Principality (also known as "Rascia") from 1166 to 1196. A member of the Vukanovic dynasty, Nemanja founded the Nemanjic dynasty, and is remembered for his contributions to Serbian culture and history, founding what would evolve into the Serbian Serbian Orthodox Church. The Serbian Orthodox Church canonized Stefan Nemanja shortly after his death under the name Saint Symeon the Myrrh-streaming after numerous miracles. Nemanja was born around the year 1113 AD in Ribnica, Zeta (in the vicinity of present-day Podgorica, the capital of Montenegro). He was the youngest son of Zavida, a Prince of Zahumlje, who after a conflict with his brothers was sent to Ribnica where he had the title of "Lord". Zavida ("Beli Uros") was most probably a son of Uros I or Vukan. Since western Zeta was under Roman Catholic jurisdiction, Nemanja received

**Output** Intermediate Answer_3: Based on the Retrieved Document_3, Stefan Nemanja died on 13 February 1199 in Hilandar monastery. Therefore, the place of death of Anastasia Of Serbia's husband is Hilandar monastery.
Final Answer: Hilandar monastery.

## C.8 CASE STUDY WITH MULTI-HOP QUESTIONS

We provide an additional case study containing multi-hop questions to help readers gain a deeper understanding of the characteristics and effectiveness of Auto-RAG.

Table 13: Question and golden answer for the case study.

**Question:** When did Edward Cromwell, 3Rd Baron Cromwell's father die?

**Golden Answer:** 20 November 1592.

Table 14: Retrieval trace for Auto-RAG.

| Step 0 |
| --- |
| **Input** When did Edward Cromwell, 3Rd Baron Cromwell's father die? |
| **Output** Analysis: To answer this question, I would need to know the name of Edward Cromwell, 3rd Baron Cromwell's father, and his death date. |
| Initial Query: Who is the father of Edward Cromwell, 3rd Baron Cromwell? |

| Step 1 |
| --- |
| **Input** Retrieved Document_1: Edward Cromwell, 3rd Baron Cromwell Edward Cromwell, 3rd Baron Cromwell (c. 1560 – 27 April 1607) was an English peer. He was the son of Henry Cromwell, 2nd Baron Cromwell by his wife Mary ...... |
| **Output** Intermediate Answer_1: Based on the Retrieved Document_1, Edward Cromwell, 3rd Baron Cromwell's father is Henry Cromwell, 2nd Baron Cromwell. To answer the question, we need to find the date of death of Henry Cromwell, 2nd Baron Cromwell. |
| Refined Query: When did Henry Cromwell, 2nd Baron Cromwell die? |

| Step 2 |
| --- |
| **Input** Retrieved Document_2: Ughtred, governor of Jersey, by whom she had two children: In 1537, three years after death of Anthony Ughtred in October 1534, she married Gregory Cromwell, the son of Thomas Cromwell and 1st Baron Cromwell ...... (Some irrelavant information) |
| **Output** Intermediate Answer_2: Based on the Retrieved Document_2, there is no information about the date of death of Henry Cromwell, 2nd Baron Cromwell. We need to refine our query and search again. |
| Refined Query: What was the date of Henry Cromwell, 2nd Baron Cromwell's death? |

| Step 3 |
| --- |
| **Input** Retrieved Document_3: Cromwell Henry Cromwell, 2nd Baron Cromwell of Oakham (before 21 May 1538 – 20 November 1592) was an English peer...... |
| **Output** Intermediate Answer_3: Based on the Retrieved Document_3, Henry Cromwell, 2nd Baron Cromwell died on 20 November 1592. |
| Final Answer: 20 November 1592. |

Table 15: Retrieval trace for FLARE.

| Step 1 |
| --- |
| **Input** <8-shot demonstrations with retrieved documents> |
| Search results ranked based on relevance in descending order: |
| [1]: Edward Cromwell, 3rd Baron Cromwell Edward Cromwell, 3rd Baron Cromwell (c. 1560 – 27 April 1607) was an English peer. He was the son of Henry Cromwell, 2nd Baron Cromwell by his wife Mary, daughter of John Paulet ... |
| [2]: Down Cathedral, on 24 September 1607. Sir Arthur Chichester, when writing of his death to the council, 29 September 1607, expressed regret at his loss... |
| Question: When did Edward Cromwell, 3Rd Baron Cromwell's father die? |
| Answer: |
| **Output** Edward Cromwell, 3rd Baron Cromwell's father was Henry Cromwell, 2nd Baron Cromwell. |

| Step 2 |
| --- |

| | |
|---|---|
| **Input** <8-shot demonstrations with retrieved documents> | |
| Search results ranked based on relevance in descending order: | |
| [1]: presided over the Restoration of 1660. Richard Cromwell subsisted in straitened circumstances after his resignation...... | |
| [2]: Oliver Cromwell Oliver Cromwell (25 April 15993 September 1658) was an English military and political leader...... | |
| Question: When did Edward Cromwell, 3Rd Baron Cromwell's father die? | |
| Answer: Edward Cromwell, 3rd Baron Cromwell's father was Henry Cromwell, 2nd Baron Cromwell. | |
| **Output** Henry Cromwell, 2nd Baron Cromwell died in 1658. | |
| **Step 3** | |
| **Input** <8-shot demonstrations with retrieved documents> | |
| Search results ranked based on relevance in descending order: | |
| [1]: Oliver Cromwell Oliver Cromwell (25 April 15993 September 1658)...... | |
| [2]: presided over the Restoration of 1660. Richard Cromwell subsisted in straitened circumstances after his resignation ...... | |
| Question: When did Edward Cromwell, 3Rd Baron Cromwell's father die? | |
| Answer: Edward Cromwell, 3rd Baron Cromwell's father was Henry Cromwell, 2nd Baron Cromwell. Henry Cromwell, 2nd Baron Cromwell died in 1658. | |
| **Output** So the answer is 1658. | |

Table 16: Retrieval trace for Self-RAG.

| | |
|---|---|
| **Question** When did Edward Cromwell, 3Rd Baron Cromwell's father die? | |
| **Whether to retrieve?** True | |
| **Generate and reflection based on each retrieved document.** | |
| **Document 1** Edward Cromwell, 3rd Baron Cromwell" Edward Cromwell, 3rd Baron Cromwell Edward Cromwell, 3rd Baron Cromwell (c. 1560 – 27 April 1607)...... | |
| **Output** [Relevant]27 April 1607[Fully supported][Utility:5] *Score: 3.88* | |
| **Document 2** ......She died before 30 November 1631. Edward Cromwell's son, Thomas visited his father's former manor of Oakham in 1631 where he partook of the ancient tradition of forfeiting a horseshoe...... | |
| **Output** [Relevant]1631[Fully supported][Utility:5] *Score: 3.63* | |
| **Document 3** "Richard Cromwell" presided over the Restoration of 1660. Richard Cromwell subsisted in straitened circumstances after his resignation...... | |
| **Output** [Relevant]1658.[Fully supported][Utility:5] *Score: 3.66* | |
| **Document 4** ...... Edward Cromwell, 3rd Baron Cromwell Edward Cromwell, 3rd Baron Cromwell (c. 1560 – 27 April 1607) was an English peer...... | |
| **Output** [Relevant]27 April 1607[Fully supported][Utility:5] ***Score: 3.89*** | |
| **Document 5** "Edward Cromwell, 3rd Baron Cromwell" Down Cathedral, on 24 September 1607. Sir Arthur Chichester, when writing of his death to the council...... | |
| **Output** [Relevant]1607[Fully supported][Utility:5] *3.72* | |
| **Final Answer** 27 April 1607. | |

**Explanation for Auto-RAG** For a complex multi-hop reasoning question, "When did Edward Cromwell, 3rd Baron Cromwell's father die?", the process unfolds as follows:

- Step 0: Auto-RAG begins by conducting retrieval planning, identifying the necessary pieces of information: (1) the identity of Edward Cromwell's father and (2) the time of his death. It then generates an initial query and decomposes the question into sub-questions, starting with: "Who is Edward Cromwell's father?"

- Step 1: From the retrieval results, Auto-RAG successfully identifies Edward Cromwell's father and formulates a new, more specific query: "When did Henry Cromwell, 2nd Baron Cromwell die?"

- Step 2: Auto-RAG observes that the retrieved documents lack the required information. Rather than fabricating an answer based on irrelevant documents, it opts to slightly adjust the query, ensuring it remains aligned with the task.
- Step 3: Auto-RAG successfully retrieves relevant documents, finds sufficient information, and terminates the iterative retrieval process, producing the final answer.

**Explanation for FLARE** In the first step, FLARE successfully identified Edward Cromwell's father. However, in the second step, due to the retrieval of irrelevant documents, FLARE generated hallucinatory responses. As a result, the third step produced an incorrect conclusion. Below are explanations of the characteristics of the FLARE method:

- **High inference overhead** FLARE employs few-shot prompting to facilitate multi-turn retrieval. The standard configuration utilizes 8-shot prompting, where each demonstration comprises two documents, one question, and a chain-of-thought response. While this setup effectively guides the model in reasoning on complex questions, it incurs significant computational overhead and increases the risk of generating hallucinations. *Auto-RAG can autonomously manage the retrieval process, achieving lower costs.*
- **Unable to refuse to answer** In the second step, due to the irrelevance of the retrieved documents, the model should have declined to provide an answer. Instead, the presence of few-shot demonstrations compelled the model to imitate the provided examples and produce a forced response, resulting in the model copying an unrelated date from the documents. *Auto-RAG is capable of rejecting irrelevant knowledge when answering questions, mitigating hallucination issues.*
- **The retrieval strategy is not sufficiently flexible** FLARE determines whether to refine its output based on the probability distribution of its responses. Nonetheless, irrelevant documents increase the likelihood of hallucinatory outputs, undermining the model's judgment. Consequently, FLARE ultimately generated a hallucinated response. *Auto-RAG employs natural language to articulate its reasoning and decision-making process, resulting in more precise decisions, enhanced interpretability, and better overall performance.*

**Explanation for Self-RAG** The core idea of Self-RAG is to independently generate responses based on multiple retrieved documents and reflect on their relevance through a reflection token, assessing whether the documents support the answer and the answer's overall utility. First, Self-RAG determines whether retrieval is necessary based on the input question. Then, for each document, Self-RAG generates a response and performs reflection, scoring each path based on the probability of extracting the reflection token. Finally, it selects the highest-scoring answer as the final result. The following are the differences between Self-RAG and Auto-RAG:

- **Self-RAG generates a response for each document, regardless of its relevance** Generating answers for all documents and selecting the most confident one as the final response may seem reasonable. However, this approach introduces unnecessary overhead and overlooks the relevance between documents. *Auto-RAG rejects irrelevant information, resulting in higher efficiency.*
- **The retrieval strategy of Self-RAG is suboptimal** Self-RAG's retrieval strategy alternates between retrieval and generation. However, when all retrieved documents are irrelevant, the model is forced to generate an answer, leading to hallucinated outputs. Subsequently, the model has no opportunity to correct the generated content, resulting in error accumulation. *In contrast, Auto-RAG is more flexible in its generation timing, depending on the availability of external knowledge. When external knowledge is unavailable, it continues retrieval rather than forcing a generation, thereby mitigating hallucination issues.*

Auto-RAG focuses on leveraging the inherent reasoning and decision-making capabilities of LLMs for iterative retrieval. Auto-RAG autonomously adjusts its retrieval strategy based on the complexity of the question and the availability of external knowledge, achieving improved results.

