# OpenReview forum: "Auto-RAG: Autonomous Retrieval-Augmented Generation for Large Language Models"
_ICLR.cc/2025/Conference — Submitted to ICLR 2025_

### Official Review · Reviewer_KJp1 · 2024-10-30

**Soundness:** 3
**Presentation:** 4
**Contribution:** 3
**Rating:** 6
**Confidence:** 5

**Summary:**

This paper introduces an autonomous iterative retrieval framework for retrieval-augmented generation. By designing appropriate prompts, the authors automatically synthesize instruction-finetuning data, which is used for the model finetuning. Experimental results validate the viability and effectiveness of the proposed framework, demonstrating performance improvements across six benchmarks.

**Strengths:**

1. This paper is well-written and well-organized.
2. The authors propose an autonomous iterative retrieval framework that, in my opinion, not only demonstrates performance improvements but also offers an efficient approach to addressing the long-context limitations of current LLMs' context windows.
3. Extensive experiments demonstrate its effectiveness, and the provided prompts used to synthesize instruction-finetuning data are valuable to this community.

**Weaknesses:**

1. Although the authors provide a detailed description of the prompts used to synthesize the dataset, the paper lacks essential information on the training process, which weakens its overall impact (see my questions for specifics).

2. The authors used Llama-3-Instruct-8B to synthesize the data and subsequently fine-tuned the same model with the generated dataset. While synthesizing data with larger models like GPT-4 or Claude would likely yield better results, the choice of an open-source model is understandable due to cost constraints. However, it raises the question of why larger Llama models, such as 70B or even 405B, weren’t used. Larger models could produce higher-quality data, potentially enhancing the training outcomes.

3. Some experimental results using GPT-4 or Claude should be provided, even if only on one dataset. While it's not expected for the finetuned Llama-3-Instruct-8B to outperform these models, such comparisons would give a clearer sense of the performance gap, better illustrating the value of fine-tuning smaller-scale language models.

**Questions:**

1. In Section 3.2, please clarify the specifics of the input $x_t$ and the output $y_t$ in Equation 3. Although $x_t$ and the output $y_t$
may seem intuitive, the synthesized datasets described in Line 17 of Algorithm 1 do not explicitly define these variables. Providing a clear description of $x_t$ and the output $y_t$ would improve understanding.

2. Could you explain what is meant by "d contains a sub-answer of X"? I wasn’t able to find the criteria used to determine this. Some examples, especially related to Natural Questions (NQ) and multi-hop question answering (HQA), would be helpful for understanding your approach.

3. In Lines 5 through 10, if the condition in Line 9 is not met after iterating over all queries in 𝑄, then $Q_k$ and $D_k$ remain undefined, which could lead to issues in Line 11. Could you clarify how this situation is handled?

4. How do you determine the value of K? Is there a risk that the context length might exceed a manageable size? If so, do you have a method for truncating context to ensure efficiency?

5. What does it mean "if no information need in R then" in Line 12 of Algorithm 1, please provide an accurate definition for that?

---

> ### Author Response · Authors · 2024-11-19
> **Response to Dear Reviewer KJp1 [1/3]**
>
> # 1. Confusion about $x_t$ and $y_t$
>
> We apologize for the confusion.
>
> Algorithm 1 primarily describes how to construct the retrieval-reasoning chain from the question to the answer, while in the "Data Formatting" paragraph in Section 3.1.2, titled "Data Filtering and Formatting," and in Equation (2), we describe how we construct input-output pairs based on the synthesized $\mathcal{D}^{inst}$. Generally, the input for each round consists of the user's question or the retrieved document, while the output is either the reasoning for that round, a query for the next retrieval (when information is insufficient), or the Final Answer (when sufficient information is available). The relevant description may not have been placed in the appropriate location due to formatting issues. We have made the necessary adjustments.
>
> Now, Equation 2 and the training section are located on page 5 and are placed adjacent to each other for easier reading.
>
> **Thank you once again for your constructive feedback.**
>
> # 2. Confusion about the expression "d contains a sub-answer of X"
>
> We apologize for the confusion.
>
> Here is an example: For the question 'What is the date of birth of Mina Gerhardsen's father?', it can be decomposed into two **sub-questions**: (1) Who is Mina Gerhardsen's father? (2) What is the date of birth of her father? The corresponding **answers** are: (1) Her father is 'Rune Gerhardsen' and (2) 'Rune Gerhardsen's date of birth is '13 June 1946'. **These two answers are considered sub-answers to the original question.** In the multi-hop question answering dataset 2WikiMultihopQA, sub-answers are provided, making the use of this information for data filtering effective.
>
> **Thank you once again for your constructive feedback.**
>
> # 3. Confusion about the exception handling method when line 9 in Algo1 is not satisfied
>
> We apologize for the confusion.
>
> If none of the sampled questions satisfy the condition in line 9, Algo 1, we will randomly select a query and its corresponding document for the next generation. We have modified Algo1 and its corresponding description to clarify this point.
>
> **Thank you once again for your constructive feedback.**
>
> # 4. Confusion regarding the maximum iteration counts
>
> We apologize for the confusion.
>
> In constructing the training data, we set the maximum number of iterations to 10 for 2WikimultifhopQA and 5 for Natural Questions. Since 2WikiMultihopQA may include questions with up to 4 hops, we provide more than two iterations for each hop. Based on our observations, if iteration does not stop within 10 steps, it is typically due to the difficulty of retrieving the relevant knowledge from the corpus, and such samples are discarded. During the construction of the training data, we use few-shot prompting, which carries a risk of exceeding the maximum length (8192 for Llama-3-8B-Instruct). Samples that exceed this length are also discarded. During testing, as few-shot prompting is no longer required, we have never observed any instances exceeding the maximum length. Indeed, our method only requires an average of 1,178 tokens on 2WikiMultiHopQA and 944 tokens on NQ, making it significantly faster than methods that utilize 8-shot with retrieved documents.
>
> **Thank you once again for your constructive feedback.** *In accordance with your suggestion, we have added the relevant information as a footnote on Page 2.*
>
> **If my response has addressed your concerns, we kindly ask that you reassess our work. We would greatly appreciate it if you could consider raising your rating!**

---

> ### Author Response · Authors · 2024-11-19
> **Response to Dear Reviewer KJp1 [2/3]**
>
> # 5. Confusion about the expression "if no information need in R then" in Algorithm 1
>
> We apologize for the confusion.
>
> We have predefined a set containing several words that indicate incomplete information, namely 'however', 'no information', 'find', and 'refine'. Therefore, the condition 'if no information is needed in R' means that none of the words in this set are present in R. We will add the appropriate explanation to improve clarity.
>
> **Thank you once again for your constructive feedback. In accordance with your suggestion, we have added the relevant information as a footnote on Page 3.**
>
>
> # 6. Considerations on Using More Powerful Large Models for Data Synthesis
>
>
> Thank you for your constructive feedback.
>
> Utilizing more powerful large models for data synthesis typically yields better results. **However, this raises new questions: is the improvement in performance attributed to the distillation of a stronger model's capabilities, or is it due to the design of our method?** To validate the effectiveness of our method, both the synthesized data and the fine-tuned models were based on the same model. In fact, our approach is similar to a self-improvement strategy, where the data generated by the model is used to imporve the model itself.
>
> **Thank you once again for your constructive feedback.**
>
> **If my response has addressed your concerns, we kindly ask that you reassess our work. We would greatly appreciate it if you could consider raising your rating!**

---

> ### Author Response · Authors · 2024-11-19
> **Response to Dear Reviewer KJp1 [3/3]**
>
> # 7. Suggestions for Supplementing GPT-4 Results
>
> Thanks for your valuable suggestions!
>
> Due to budget and time constraints, we sampled 1,000 samples from each dataset and compared the performance of our method with that of closed-source models. The random seed was set to 0.
> ```
> +--------------+------------------------+-------+-------+----------+-------+----------+-------+-------+
> | Method       | Model                  |   NQ  | 2Wiki | TriviaQA | PopQA | HotpotQA | WebQA | AVG   |
> +--------------+------------------------+-------+-------+----------+-------+----------+-------+-------+
> |                                             No Retrieval                                            |
> +--------------+------------------------+-------+-------+----------+-------+----------+-------+-------+
> | Naive-Gen    | gpt-4o-2024-08-06      |  16.9 | 43.2  |   69.1   | 48.2  |   48.3   | 15.1  | 40.1  |
> |              +------------------------+-------+-------+----------+-------+----------+-------+-------+
> |              | gpt-4o-mini-2024-07-18 |  19.2 | 31.7  |   59.6   | 35.1  |   37.9   | 19.9  | 33.9  |
> |              +------------------------+-------+-------+----------+-------+----------+-------+-------+
> |              | Llama-3-8B-Instruct    |  20.9 | 25.7  |   54.0   | 26.3  |   27.1   | 20.1  | 29.0  |
> +--------------+------------------------+-------+-------+----------+-------+----------+-------+-------+
> |                                          Standard Retrieval                                         |
> +--------------+------------------------+-------+-------+----------+-------+----------+-------+-------+
> | Standard RAG | gpt-4o-2024-08-06      |  14.0 | 36.2  |   58.7   | 45.6  |   46.8   | 13.9  | 35.9  |
> |              +------------------------+-------+-------+----------+-------+----------+-------+-------+
> |              | gpt-4o-mini-2024-07-18 |  29.9 | 34.0  |   61.3   | 49.6  |   45.6   | 19.5  | 40.0  |
> |              +------------------------+-------+-------+----------+-------+----------+-------+-------+
> |              | Llama-3-8B-Instruct    |  35.1 | 19.1  |   56.9   | 47.5  |   35.6   | 16.3  | 35.1  |
> +--------------+------------------------+-------+-------+----------+-------+----------+-------+-------+
> |                                         Autonomous Retrieval                                        |
> +--------------+------------------------+-------+-------+----------+-------+----------+-------+-------+
> | Auto-RAG     | Llama-3-8B-Instruct    | 34.2  | 47.9  |   58.6   | 48.4  |   45.7   | 23.4  | 43.0  |
> +--------------+------------------------+-------+-------+----------+-------+----------+-------+-------+
> ```
>
>
> The following presents the conclusions of the experimental results:
>
> 1. The average performance of Auto-RAG is the best.
> 2. GPT-4o demonstrated better performance without retrieval, while GPT-4o-mini showed improved performance after retrieval.
> 3. For a well-trained model, the quality of its parameterized knowledge may surpass that of external knowledge, so providing external knowledge may degrade performance. **Enhancing the model's ability to resist irrelevant information is crucial.**
>
> **Auto-RAG autonomously adjusts its retrieval strategy based on the availability of external knowledge. When external knowledge is useful, it answers sub-questions, generates new queries, or derives the final conclusion. If the external knowledge is not useful, it refuses to answer and re-initiates the search process.** This way, the model is not forced to generate hallucinations, thereby improving the quality of the output.
>
> **Thank you once again for your constructive feedback.** *Due to space limitations and the fact that the test is based on a sample of 1,000 data points, which differs from the main experiment, we will provide additional details in the Appendix A2, Page 17.*
>
> **If my response has addressed your concerns, we kindly ask that you reassess our work. We would greatly appreciate it if you could consider raising your rating!**

---

> ### Author Response · Authors · 2024-12-03
> **Kind Reviewer for Dear Reviewer KJp1**
>
> Dear Reviewer KJp1:
>
> We sincerely thank you for your hard work during the review and rebuttal process. We have carefully read your comments, provided additional clarifications, and made revisions to the paper. We are more than happy to engage in discussions with you, as it plays a crucial role in resolving your concerns and improving our paper!
>
> **However, as the rebuttal period will conclude in two days, we would be grateful if you could let us know whether we have addressed your concerns at your earliest convenience. If you still have any concerns, we would be delighted to address them promptly.  If your concerns have been addressed, we would appreciate it if you could kindly reassess our work and increase the rating.**
>
> Best regards,
>
> Authors

---

### Official Review · Reviewer_b21a · 2024-10-31

**Soundness:** 3
**Presentation:** 3
**Contribution:** 2
**Rating:** 5
**Confidence:** 4

**Summary:**

This paper presents Auto-RAG, an autonomous iterative retrieval model that engages in multi-turn dialogues with the retriever. It systematically plans its retrievals and refines queries to gather valuable knowledge. In addition, they develop a method for autonomously synthesizing reasoning-based decision-making instructions in iterative retrieval and fine-tuned the latest open-source LLMs. The experimental results show that Auto-RAG can engage in autonomous iterative interactions with the retriever, effectively utilizing the impressive reasoning and decision-making capabilities of large language models, resulting in excellent performance across six benchmarks.

**Strengths:**

(1)They have conducted thorough research and organization of the work in the area of iterative retrieval within the RAG field.
(2)They proposed an effective data synthesis method utilizing large language models specifically for this direction, enabling fine-tuning based on this approach.
(3)The entire Auto-RAG process is efficient, with clear and coherent language expression throughout.

**Weaknesses:**

(1)The contribution of “parametric knowledge” to the overall reasoning process is not clearly articulated. Could you provide concrete examples of how parametric knowledge influences the model's reasoning or decision-making?
(2)The overall approach does not seem to present significant innovations compared to previous work, such as self-rag; it primarily enriches the prompt content.
(3)The whole pipeline relies on the presence of the word "answer" in the generated content to determine when to conclude the retrieval process. This method may be susceptible to the model's "hallucinations."

**Questions:**

(1)In the inference phase of Auto-RAG, why is the utilization of parametric knowledge placed later in the process? Would it not be more beneficial for overall efficiency to incorporate it before retrieval? Additionally, further analysis is needed to quantify the contribution of parametric knowledge to the final generation. Could you provide empirical evidence or reasoning for why they chose this particular ordering of parametric knowledge utilization?
(2)In the results presented in Table 1, why does Naive Gen perform better on 2Wiki than nearly all the baselines? Is it possible that the baselines were not provided with appropriately configured prompts to elicit suitable answers?
(3)Why are the performance metrics for Iter-RetGen not displayed in Figure 7?

---

> ### Author Response · Authors · 2024-11-19
> **Response to Dear Reviewer b21a [1/2]**
>
> # 1. Questions Regarding the Order of External Knowledge and Parameterized Knowledge Provided to Auto-RAG
>
> Thanks for your valuable suggestions!
>
> **First, we will briefly explain the rationale behind this setup. Subsequently, we will present additional results to demonstrate its effectiveness.**
>
> External knowledge refers to the information retrieved by the retriever from the knowledge base based on the query. Parametric knowledge refers to the documents generated by the LLMs based on the input query. The reason we first provide external knowledge to the model and then parametric knowledge is as follows:
> 1. As shown in the main experiment in Table 1, the model performs better on average when external knowledge is provided (Standard RAG vs Naive Gen). This suggests that, for LLaMA 3-8B, external knowledge may be more valuable.
> 2. The knowledge generated by the model is **highly misleading** (Sec. 4.1 in [1]). The model is capable of generating more coherent yet fabricated knowledge that is convincing to LLMs.
>
> Therefore, we first provide external knowledge, followed by parameter-based knowledge. When external knowledge is irrelevant, the model is able to reject the answer and perform further retrieval. However, when provided with parametric knowledge generated by LLMs, it becomes difficult for the model to escape this trap, ultimately leading to hallucinated outputs.
>
> References:
> [1] [Adaptive Chameleon or Stubborn Sloth: Revealing the Behavior of Large Language Models in Knowledge Conflicts](https://arxiv.org/pdf/2305.13300).
>
> Next, we present additional results to validate the effectiveness of this design.
>
> We swapped the order of providing external knowledge and parameter-based knowledge, keeping all other settings unchanged. The experimental results are as follows:
>
> ```
> +---------------------+-------+-------+----------+----------+-------+-------+-------+
> |                     |   NQ  | 2Wiki | TriviaQA | HotpotQA | PopQA |  WebQ |       |
> | Order               +-------+-------+----------+----------+-------+-------+  AVG  |
> |                     |   EM  |   F1  |    EM    |    F1    |  F1   |   EM  |       |
> +---------------------+-------+-------+----------+----------+-------+-------+-------+
> | parametric-external | 26.7  |  37.4 |   54.3   |   33.8   | 34.6  |  18.2 | 34.2  |
> +---------------------+-------+-------+----------+----------+-------+-------+-------+
> | external-parametric | 37.9  |  48.9 |   60.9   |   47.8   | 44.9  |  25.1 | 44.3  |
> +---------------------+-------+-------+----------+----------+-------+-------+-------+
> ```
>
> The results indicate that providing external knowledge first indeed leads to better performance.
>
> To further validate Reason 2, we calculated the proportion of iterations for both sequences on the NQ dataset:
>
> ```
> +---------------------+-------------------------------------------------+
> |                     |        Distributions of Iteration Counts        |
> | Order               +--------+--------+-------+-------+-------+-------+
> |                     |    1   |    2   |   3   |   4   |   5   |   6   |
> +---------------------+--------+--------+-------+-------+-------+-------+
> | parametric-external | 82.08% | 8.98%  | 0.50% | 0.30% | 0.28% | 6.70% |
> +---------------------+--------+--------+-------+-------+-------+-------+
> | external-parametric | 44.65% | 47.56% | 2.94% | 0.97% | 0.58% | 2.33% |
> +---------------------+--------+--------+-------+-------+-------+-------+
> ```
>
> When parameter-based knowledge is provided first, Auto-RAG performs fewer iterations. This demonstrates that the content generated by LLMs is misleading and convincing, consistent with the findings in [1].
>
>
> **Thank you once again for your constructive feedback.** *Due to space limitations, we will present this interesting experiment in the Apendix A3, Page 17-18.*
>
> **If my response has addressed your concerns, we kindly ask that you reassess our work. We would greatly appreciate it if you could consider raising your rating!**

---

> > ### Author Response · Authors · 2024-11-25
> > **Additional Results for Dear Reviewer b21a**
> >
> > # Additional Results Regarding the Impact of External Knowledge
> >
> > To further investigate the impact of parametric knowledge on Auto-RAG's performance, we conducted an additional experiment with only external knowledge (no-parametric). The results of the new experiment are shown in the table below:
> >
> > ```
> > +---------------------+-------+-------+----------+----------+-------+-------+-------+
> > | Order               |   NQ  | 2Wiki | TriviaQA | HotpotQA | PopQA |  WebQ |  AVG  |
> > |                     +-------+-------+----------+----------+-------+-------+       |
> > |                     |   EM  |   F1  |    EM    |    F1    |   F1  |   EM  |       |
> > +---------------------+-------+-------+----------+----------+-------+-------+-------+
> > | no-parametric       | 37.7  | 39.8  |   60.1   |   42.0   | 46.9  | 22.6  | 41.5  |
> > +---------------------+-------+-------+----------+----------+-------+-------+-------+
> > | parametric-external | 26.7  | 37.4  |   54.3   |   33.8   | 34.6  | 18.2  | 34.2  |
> > +---------------------+-------+-------+----------+----------+-------+-------+-------+
> > | external-parametric | 37.9  | 48.9  |   60.9   |   47.8   | 44.9  | 25.1  | 44.3  |
> > +---------------------+-------+-------+----------+----------+-------+-------+-------+
> > ```
> >
> > The experimental results (no-parametric vs. external-parametric) show that using only external knowledge yields good performance, and supplementing with parametric knowledge further enhances the results.
> >
> > We believe that our experiments effectively address your concerns. Thank you again for your constructive feedback, which has helped improve our work.
> >
> > **If my response has addressed your concerns, we kindly ask that you reassess our work. We would greatly appreciate it if you could consider raising your rating!**

---

> ### Author Response · Authors · 2024-11-19
> **Response to Dear Reviewer b21a [2/2]**
>
> # 2. Confusions about the Results on 2WikiMultihopQA
>
> We apologize for the confusion.
>
> We use the prompts officially provided by FlashRAG, as shown in Appendix C2, Page 22. WikiMultihopQA is a challenging dataset (with many examples requiring up to 4 hops [1]). When the number of provided documents is limited (5, consistent with the official settings [2]), it is difficult to obtain sufficient knowledge from external sources, and the resulting decline in performance is expected.
>
> References:
>
> [1] [Constructing A Multi-hop QA Dataset for Comprehensive Evaluation of Reasoning Steps](https://aclanthology.org/2020.coling-main.580/)
>
> [2] [FlashRAG: A Python Toolkit for Efficient RAG Research](https://arxiv.org/abs/2405.13576)
>
> # 3. Regarding the absence of Iter-RetGen in Figure 7
>
> Thanks for your valuable feedback!
>
> In fact, the number of iterations for Iter-RetGen is entirely controlled manually, making it unsuitable for time-based analysis.
>
> **Thank you once again for your valuable feedback.**
>
> References:
>
> [1] [Enhancing Retrieval-Augmented Large Language Models with Iterative Retrieval-Generation Synergy](https://arxiv.org/abs/2305.15294)
>
> # 4. Concerns regarding the contributions of this paper
>
> Thanks for your valuable feedback!
>
> We will further clarify the contributions of our approach to facilitate your understanding of our innovations.
>
> In this paper, **we focus on leveraging the reasoning and decision-making capabilities of LLMs for iterative retrieval, introducing Auto-RAG**. The core of iterative retrieval lies in the design of the retrieval strategy (determining the timing and content of the retrieval). However, existing approaches often rely on few-shot prompting and manually constructed rules to implement iterative retrieval. For example, Self-RAG follows a strict retrieval-generation alternation process and uses reflection tokens to reflect on its output [1]; FLARE generates first and then refines, deciding whether additional retrieval is needed based on output probabilities [2]; Iter-RetGen relies entirely on a manually set number of retrieval steps, with each query being the output from the previous step [3]. Although these methods have shown some success, they are inefficient and fail to fully harness the reasoning and decision-making abilities of LLMs. Auto-RAG offers the following notable advantages:
>
> * **More flexible retrieval strategy.** Unlike the strict retrieval-generation alternation, our approach generates when knowledge is sufficient and continues retrieval when knowledge is lacking. This more flexible approach helps avoid hallucinations caused by forced generation when information is insufficient.
> * **Higher interpretability.** Auto-RAG fully articulates its reasoning and decision-making process in natural language. Incorporating reasoning into the retrieval process enables more precise decision-making (See Ablation Study).
> * **Full utilization of LLMs' reasoning capabilities.** Auto-RAG adapts the number of iterations based on the availability of external knowledge and the difficulty of the question, improving performance across various datasets and demonstrating strong robustness.
>
> References:
>
> [1] [Self-RAG: Learning to Retrieve, Generate, and Critique through Self-Reflection](https://arxiv.org/abs/2310.11511)
>
> [2] [Active Retrieval Augmented Generation](https://arxiv.org/abs/2305.06983)
>
> [3] [Enhancing Retrieval-Augmented Large Language Models with Iterative Retrieval-Generation Synergy](https://arxiv.org/abs/2305.15294)
>
> **Thank you once again for your valuable feedback.**
>
> # 5. Concerns regarding the impact of hallucinations
>
> Thanks for your valuable feedback!
>
> In fact, hallucinations are an inherent challenge. To mitigate this, we integrate reasoning into the iterative retrieval process, enabling the model to better understand the rationale behind each decision. When knowledge is insufficient, Auto-RAG avoids forced generation and instead continues retrieval. **This flexible retrieval strategy has significantly alleviated hallucination issues.**
>
> **Thank you once again for your valuable feedback.**
>
> **If my response has addressed your concerns, we kindly ask that you reassess our work. We would greatly appreciate it if you could consider raising your rating!**

---

> ### Author Response · Authors · 2024-12-03
> **Kind Reminder for Dear Reviewer b21a**
>
> Dear Reviewer b21a:
>
> We sincerely thank you for your hard work during the review and rebuttal process. We have carefully read your comments, provided additional clarifications, and made revisions to the paper. We are more than happy to engage in discussions with you, as it plays a crucial role in resolving your concerns and improving our paper!
>
> **However, as the rebuttal period will conclude in two days, we would be grateful if you could let us know whether we have addressed your concerns at your earliest convenience. If you still have any concerns, we would be delighted to address them promptly.  If your concerns have been addressed, we would appreciate it if you could kindly reassess our work and increase the rating.**
>
> Best regards,
>
> Authors

---

### Official Review · Reviewer_eSB7 · 2024-10-31

**Soundness:** 3
**Presentation:** 2
**Contribution:** 3
**Rating:** 5
**Confidence:** 4

**Summary:**

This paper introduces Auto-RAG, an autonomous iterative retrieval model leveraging large language models’ (LLM) decision-making capabilities. Auto-RAG engages in multi-turn dialogues with the retriever, methodically planning and refining queries to acquire necessary external knowledge. This iterative process continues until sufficient information is gathered, at which point the results are provided to the user. The authors propose a method to autonomously generate reasoning-based decision-making instructions in iterative retrieval and fine-tune the latest open-source LLMs to enhance performance.

**Strengths:**

- The model effectively adapts iteration count for optimized performance.
- This work opens a promising new direction in automating retrieval-augmented generation (RAG).

**Weaknesses:**

- Novelty seems to  limited, as automated iteration in prompt-based systems has been explored in other works, including ReACT, Tool-LLM, and Chameleon (https://arxiv.org/abs/2304.09842). The approach to fine-tuning also appears similar to Alpaca.

- Critical baselines are absent, such as BeamAGGR (https://arxiv.org/pdf/2406.19820), Self-CRAG, and Open-RAG (https://arxiv.org/abs/2410.01782)—though the latter is concurrent and could be considered optional.

- Some reported results raise questions. For instance, on the TQA dataset, Self-RAG reports a 66+ accuracy using Llama 2, yet Auto-RAG with Llama 3.1 only achieves 38, while the baseline is 55.7. Similar discrepancies are noted for other datasets, like PQA.

- Section 5.5’s evaluation method of Self-RAG is unclear. Although these models say they process top-k retrievals in parallel but for arguments even if we consider the processing is individually to generate multiple answers, speed should theoretically scale with the number of retrievals. Figure 7 is therefore confusing: why does Self-RAG demonstrate slower speeds with fewer retrievals (e.g., in 2 Wiki)?

- Algorithms 1 and 2 are insufficiently clear. For example, in Algorithm 1, line 9’s method for determining if d contains a sub-answer of X is ambiguous. Additionally, the writing from line 216 onward does not consistently align with the notations in Algorithms 1 and 2 (e.g., maximum iterations are denoted as T vs. K). Improving and synchronizing these algorithmic and training details would benefit clarity.

**Questions:**

See weaknesses.

---

> ### Author Response · Authors · 2024-11-19
> **Response to Dear Reviewer eSB7 [1/5]**
>
> **We sincerely thank the reviewer for the constructive and helpful feedback. We hope the following responses would help address your concerns.**
>
> # 1. Considerations Regarding the Contributions of Auto-RAG
>
> Thanks for your valuable feedback!
>
> We will further clarify the contributions of our approach to facilitate your understanding of our innovations.
>
> In this paper, **we focus on leveraging the reasoning and decision-making capabilities of LLMs for iterative retrieval, introducing Auto-RAG**. The core of iterative retrieval lies in the design of the retrieval strategy (determining the timing and content of the retrieval). However, existing approaches often rely on few-shot prompting and manually constructed rules to implement iterative retrieval. For example, Self-RAG follows a strict retrieval-generation alternation process and uses reflection tokens to reflect on its output [1]; FLARE generates first and then refines, deciding whether additional retrieval is needed based on output probabilities [2]; Iter-RetGen relies entirely on a manually set number of retrieval steps, with each query being the output from the previous step [3]. Although these methods have shown some success, they are inefficient and fail to fully harness the reasoning and decision-making abilities of LLMs. Auto-RAG offers the following notable advantages:
>
> * **More flexible retrieval strategy.** Unlike the strict retrieval-generation alternation, our approach generates when knowledge is sufficient and continues retrieval when knowledge is lacking. This more flexible approach helps avoid hallucinations caused by forced generation when information is insufficient.
> * **Higher interpretability.** Auto-RAG fully articulates its reasoning and decision-making process in natural language. Incorporating reasoning into the retrieval process enables more precise decision-making (See Ablation Study).
> * **Full utilization of LLMs' reasoning capabilities.** Auto-RAG adapts the number of iterations based on the availability of external knowledge and the difficulty of the question, improving performance across various datasets and demonstrating strong robustness.
>
> To empower LLMs with autonomous decision-making capabilities in iterative retrieval at a minimal cost, we develop a method for autonomously synthesizing reasoning-based decision-making instructions in iterative retrieval and fine-tuned the latest open-source LLMs. We employ standard supervised fine-tuning (SFT) to train the model, as it is widely used, effective, and sufficient for our purposes, given that the training methodology itself is not the core focus of our study.
>
> References:
>
> [1] [Self-RAG: Learning to Retrieve, Generate, and Critique through Self-Reflection](https://arxiv.org/abs/2310.11511)
>
> [2] [Active Retrieval Augmented Generation](https://arxiv.org/abs/2305.06983)
>
> [3] [Enhancing Retrieval-Augmented Large Language Models with Iterative Retrieval-Generation Synergy](https://arxiv.org/abs/2305.15294)
>
> **Thank you once again for your valuable feedback.**
>
> **Dear reviewer eSB7, your feedback is of utmost importance to us. We hope our response has addressed your concerns, and if you have any further questions, please feel free to continue the discussion. We would greatly appreciate it if you could reassess our work and raise your score.**

---

> ### Author Response · Authors · 2024-11-19
> **Response to Dear Reviewer eSB7 [2/5]**
>
> # 2. Suggestions for Additional Baselines
>
> Thanks for your valuable suggestions!
>
> Following your suggestions, we conducted a systematic review of the BeamAGGR and Self-CRAG papers.
>
> BeamAGGR is a method designed for multi-hop reasoning. This approach begins by decomposing complex questions through few-shot prompting (Sec. 3.1, Page 4). It then addresses the subproblems using knowledge from multiple sources, including parametric knowledge, Wikipedia, and web search (Sec. 3.2, Page 4). Finally, it adopts a bottom-up strategy to identify the optimal reasoning path (Sec. 3.3, Page 4). This approach has demonstrated strong performance on several multi-hop question answering datasets.
> **However, for the following reasons, a direct comparison with our method is not feasible:**
> * **The core focus of the research differs.** BeamAGGR is designed to address multi-hop problems, focusing on question decomposition and answer fusion. In contrast, Auto-RAG targets more general open-domain question answering, with an emphasis on developing retrieval strategies for iterative processes. It leverages the reasoning and decision-making capabilities inherent in LLMs to enable more flexible and efficient decision-making.
> * The backbone models are different. This method is based on the powerful proprietary model GPT-3.5-turbo.
> Our method is based on the latest open-source model, Llama-3 series.
> * The scope of testing differs. Due to time and budget constraints, they randomly selected only 500 test samples for evaluation. Auto-RAG is not constrained by budget, allowing us to test it on the entire test set.
> * The sources of knowledge differ. Their method integrates web search, which demonstrates superior performance compared to local retrieval.
>
> Self-CRAG's core idea is to supplement knowledge through web retrieval and revise previous answers when local retrieval is deemed unhelpful. This approach alleviates the hallucination problem caused by insufficient knowledge in the local knowledge base.
> **However, for the following reasons, a direct comparison with our method is not feasible:**
> * **The core focus of the research differs.** Self-CRAG focus on performing post-correction on the generated content, primarily relying on supplementary web retrieval to address the issue of insufficient knowledge from the local knowledge source. In contrast, Auto-RAG is primarily concerned with the design of the retrieval strategy.
> * The sources of knowledge differ. Self-CRAG utilizes web retrieval (Sec. 4.5, Page 6 in Arxiv Preprint of CRAG) while Auto-RAG uses local search.
> * The scope of testing differs. This method selects a subset of the PopQA test set, whereas we use the entire test set (Appendix B1 in Arxiv Preprint of CRAG).
> * The evaluation metrics differ between the methods. This approach uses Accuracy, where the final output is considered correct if any golden answer appears. In contrast, our metrics are F1 and Exact Match (EM).
>
> **Therefore, a direct comparison between these methods would be unfair.**
>
> **Thank you once again for your constructive feedback.** *Although we are unable to directly compare with these baselines, we will include them in our Related Works.*
>
> **Dear reviewer eSB7, your feedback is of utmost importance to us. We hope our response has addressed your concerns, and if you have any further questions, please feel free to continue the discussion. We would greatly appreciate it if you could reassess our work and raise your score.**

---

> ### Author Response · Authors · 2024-11-19
> **Response to Dear Reviewer eSB7 [3/5]**
>
> # 3. Confusion Regarding Self-RAG's Results
>
> Thanks for your valuable feedback!
>
> The discrepancies between the results of Self-RAG in our study and those reported in the original paper can be attributed to the following reasons:
> * **The evaluation metrics differ.** The Self-RAG paper reports results based on Accuracy, where a sample is considered correct if its output contains any golden answer (paragraph 2, page 7 in Self-RAG paper). This metric is more lenient compared to the stricter metrics reported in our paper, namely Exact Match (EM) and F1.
> * **The scope of the test set differs.** Self-RAG's evaluation on PopQA is conducted on a subset of the test set (paragraph 2, page 7 in Self-RAG paper) rather than the entire test set.
>
> To ensure a fair comparison, we also evaluated Auto-RAG on the test set used in their study, applying the same metric. If the Final Answer generated by Auto-RAG contains any of the golden answers, it is considered correct. The results are summarized in the table below:
>
>
> ```
> +----------+----------+-------+
> |          | TriviaQA | PopQA |
> | Method   +----------+-------+
> |          |    Acc   |  Acc  |
> +----------+----------+-------+
> | Self-RAG |   69.3   |  55.8 |
> +----------+----------+-------+
> | Auto-RAG |   70.2   |  59.7 |
> +----------+----------+-------+
> ```
>
>
> The results for Self-RAG are derived from the original study. **Auto-RAG performs better than Self-RAG.** Notably, Self-RAG utilizes GPT-4 to synthesize data for training its critique model, whereas we rely solely on open-source models for data synthesis. Furthermore, the training data for Self-RAG comprises 150K examples, while Auto-RAG is trained on only 10K examples.
>
> *As the experimental setup of Self-RAG in the original study differs from that of our main experiments, we present these results in Appendix A4, page18.*
>
> **Thank you once again for your valuable feedback.**
>
> **Dear reviewer eSB7, your feedback is of utmost importance to us. We hope our response has addressed your concerns, and if you have any further questions, please feel free to continue the discussion. We would greatly appreciate it if you could reassess our work and raise your score.**

---

> ### Author Response · Authors · 2024-11-19
> **Response to Dear Reviewer eSB7 [4/5]**
>
> # 4. Confusion Regarding Self-RAG Speed Evaluation
>
> Thanks for your valuable feedback!
>
> We deployed Self-RAG using the official implementation of FlashRAG. **The methods and args for deploying both the retriever and the model are identical to those used in Auto-RAG.** However, we observed that Self-RAG requires the output probabilities and  rule-based complex calculations on the extracted probabilities, the time overhead increases significantly, resulting in longer runtime for Self-RAG.
>
> **Thank you once again for your valuable feedback.**
>
> # 5 Confusion Regarding Algorithmic Descriptions
>
> We apologize for the confusion.
>
> Here, we provide an explanation for the comment "the method for determining if d contains a sub-answer of X in Algorithm 1, line 9, is ambiguous." As stated on lines 230-232 of the paper, for multi-hop reasoning datasets that provide sub-answers, we filter the generated queries. Each generated query is used for retrieval, and queries that retrieve documents containing a sub-answer are retained.
>
> For example, the question 'What is the date of birth of Mina Gerhardsen's father?' can be decomposed into two **sub-questions**: (1) Who is Mina Gerhardsen's father? (2) What is the date of birth of her father? The corresponding **answers** are: (1) Her father is 'Rune Gerhardsen' and (2) 'Rune Gerhardsen's date of birth is '13 June 1946'. **These two answers are considered sub-answers to the original question.** If the document retrieved by query q contains any sub-answer, q will be retained.
>
> Additionally, in accordance with your suggestions, we have revised Algorithm 1, Equation 1, and the related description (line 159-188), standardizing the maximum number of iterations as $T$.
>
> **Thank you once again for your valuable feedback.**
>
> **Dear reviewer eSB7, your feedback is of utmost importance to us. We hope our response has addressed your concerns, and if you have any further questions, please feel free to continue the discussion. We would greatly appreciate it if you could reassess our work and raise your score.**

---

> ### Author Response · Authors · 2024-11-19
> **Response to Dear Reviewer eSB7 [5/5]**
>
> # Additional Results for Dear Reviewer eSB7
>
> We compared the performance of Auto-RAG with that of closed-source models (such as GPT-4o), hoping that these results will provide you with a deeper understanding of our contributions.
>
> Due to budget and time constraints, we sampled 1,000 samples from each dataset and compared the performance of our method with that of closed-source models. The random seed was set to 0.
>
> ```
> +--------------+------------------------+-------+-------+----------+-------+----------+-------+-------+
> | Method       | Model                  |   NQ  | 2Wiki | TriviaQA | PopQA | HotpotQA | WebQA | AVG   |
> +--------------+------------------------+-------+-------+----------+-------+----------+-------+-------+
> |                                             No Retrieval                                            |
> +--------------+------------------------+-------+-------+----------+-------+----------+-------+-------+
> | Naive-Gen    | gpt-4o-2024-08-06      |  16.9 | 43.2  |   69.1   | 48.2  |   48.3   | 15.1  | 40.1  |
> |              +------------------------+-------+-------+----------+-------+----------+-------+-------+
> |              | gpt-4o-mini-2024-07-18 |  19.2 | 31.7  |   59.6   | 35.1  |   37.9   | 19.9  | 33.9  |
> |              +------------------------+-------+-------+----------+-------+----------+-------+-------+
> |              | Llama-3-8B-Instruct    |  20.9 | 25.7  |   54.0   | 26.3  |   27.1   | 20.1  | 29.0  |
> +--------------+------------------------+-------+-------+----------+-------+----------+-------+-------+
> |                                          Standard Retrieval                                         |
> +--------------+------------------------+-------+-------+----------+-------+----------+-------+-------+
> | Standard RAG | gpt-4o-2024-08-06      |  14.0 | 36.2  |   58.7   | 45.6  |   46.8   | 13.9  | 35.9  |
> |              +------------------------+-------+-------+----------+-------+----------+-------+-------+
> |              | gpt-4o-mini-2024-07-18 |  29.9 | 34.0  |   61.3   | 49.6  |   45.6   | 19.5  | 40.0  |
> |              +------------------------+-------+-------+----------+-------+----------+-------+-------+
> |              | Llama-3-8B-Instruct    |  35.1 | 19.1  |   56.9   | 47.5  |   35.6   | 16.3  | 35.1  |
> +--------------+------------------------+-------+-------+----------+-------+----------+-------+-------+
> |                                         Autonomous Retrieval                                        |
> +--------------+------------------------+-------+-------+----------+-------+----------+-------+-------+
> | Auto-RAG     | Llama-3-8B-Instruct    | 34.2  | 47.9  |   58.6   | 48.4  |   45.7   | 23.4  | 43.0  |
> +--------------+------------------------+-------+-------+----------+-------+----------+-------+-------+
> ```
>
>
> The following presents the conclusions of the experimental results:
>
> 1. The average performance of Auto-RAG is the best.
> 2. GPT-4o demonstrated better performance without retrieval, while GPT-4o-mini showed improved performance after retrieval.
> 3. For a well-trained model, the quality of its parameter knowledge can be compromised by the quality of external knowledge. Therefore, providing external knowledge may degrade its performance. Enhancing the model's ability to resist irrelevant information is crucial.
>
> **Auto-RAG autonomously adjusts its retrieval strategy based on the availability of external knowledge. When external knowledge is useful, it answers sub-questions, generates new queries, or derives the final conclusion. If the external knowledge is not useful, it refuses to answer and re-initiates the search process.**
>
> The results of this experiment are presented in Appendix A2, Page 17. We hope that the provided results will assist you in better evaluating our work.
>
> **Dear reviewer eSB7, your feedback is of utmost importance to us. We hope our response has addressed your concerns, and if you have any further questions, please feel free to continue the discussion. We would greatly appreciate it if you could reassess our work and raise your score.**

---

> ### Comment · Reviewer_eSB7 · 2024-11-27
> **Response to Authors**
>
> Thanks for the resposne.
>
> 1. it also is required to know the baseline LLama performance in terms of new data split and metrics.
>
> 2. The distinction from the related works does not seem to be a significant novelty.
> 3. I think that you are also decomposing questions but maybe in a different way than BeamAGGR. why can't we compare with them? They are also doing the same task but in a different way. Is there any reason for that like they use fine-tuning and you are only using inference-only methods or so?
>
> Overall, I would like to maintain my score.

---

> > ### Author Response · Authors · 2024-11-27
> > **Response to Dear Reviewer eSB7**
> >
> > Thank you for your response!  We will provide further clarification on your concerns.
> >
> > # 1. Concerns about the testing scope of the Self-RAG paper
> >
> > It is possible that you may have overlooked the details of the relevant implementation in Self-RAG. We would like to point out the original text and its source for your reference. The version we referenced is https://arxiv.org/pdf/2310.11511.
> >
> > The following statement appears in the 2nd paragraph on page 7: For PopQA, we use the long-tail subset, consisting of 1,399 rare entity queries whose monthly Wikipedia page views are less than 100. (This sentence is copied from the origin paper, please kindly take note) .
> >
> > Therefore, the testing scope of self-rag differs from ours. As mentioned previously, we have conducted comparisons under the same testing scope and metrics, and our approach shows superior performance. *This part of the experiment has been included in Appendix A4, on page 19.* In subsequent versions of the paper, we will provide a more detailed clarification on this matter.
> >
> > According to your request, we have included the experimental results of Llama-3-8B-Instruct, which is shown below:
> >
> > ```
> > +-----------+----------+-------+
> > | Method    | TriviaQA | PopQA |
> > +-----------+----------+-------+
> > | Llama-3-8B|   59.3   |  19.1 |
> > +-----------+----------+-------+
> > | Self-RAG  |   69.3   |  55.8 |
> > +-----------+----------+-------+
> > | Auto-RAG  |   70.2   |  59.7 |
> > +-----------+----------+-------+
> > ```
> >
> > We will include this set of experiments in the paper. Thank you for your efforts in helping to improve our work.
> >
> > # 2. The concerns regarding the contributions of this paper
> > Auto-RAG offers a more reasonable iterative retrieval strategy. We fully leverage the inference decision-making capabilities of LLMs, adjusting the number of iterations based on the availability of external knowledge and the difficulty of the task. Specifically,  Auto-RAG determines the timing of retrieval based on the usefulness of the external knowledge. When external knowledge is deemed unuseful, Auto-RAG continues the retrieval process rather than forcing the model to generate and then correct its output.
> >
> > In contrast, FLARE determines the timing of retrieval based on whether the probability of the current output falls below a threshold, rather than the usefulness of external knowledge. However, incorrect external knowledge can also increase the probability of erroneous outputs, leading to decision failure. Self-RAG generates an answer for each document and selects the best one. When the retrieved documents are irrelevant, retrieval cannot continue, and the generated answer cannot be corrected. Iter-RetGen fully relies on manual adjustment of the number of retrieval iterations, with each retrieval being based on the output from the previous iteration, making it less flexible.
> >
> > # 3. Reasons for the inability to directly compare with BeamAGGR.
> >
> > We would like to reiterate why we are unable to conduct a comparison with BeamAGGR.
> > * First, BeamAGGR has not released its code and is built based on APIs. Its test samples are randomly sampled 500 data points (see Sections 4.1 and 4.2 on page 1233 of https://aclanthology.org/2024.acl-long.67.pdf), resulting in a potentially high degree of randomness in their testing. **Although we are eager to compare with BeamAGGR, we are unable to reproduce and test it on the same base model and test scope.**
> > * Moreover, a key focus of BeamAGGR is on how to aggregate knowledge from different sources. *Without web retrieval, the performance of their method significantly declines (see Table 3 on page 1235). Using the open-source foundation model also results in a significant performance drop (Page 1235, Table 2).*
> > * Additionally, BeamAGGR requires around 20k–30k tokens to operate (see Figure 7 on page 1236), whereas our method requires only an average of 1178 tokens on 2WikiMultiHopQA and 944 tokens on NQ. Our method is based on open-source models and significantly reduces inference costs and response latency.
> >
> > Based on the above, we believe that it is fair to compare models using the same foundation, test set, and knowledge source, although our focus may differ. As BeamARRG does not provide its code, we are unable to conduct the corresponding experiments. We kindly ask for your understanding. We would be pleased to introduce this work in the related work.
> >
> > **Thank you once again for taking the time to read our response. If the above replies have not fully addressed your concerns, we kindly ask you to continue the discussion. We would be more than happy to continue the discussion.**

---

> > > ### Comment · Reviewer_eSB7 · 2024-12-01
> > > **Response to Authors**
> > >
> > > Thanks,
> > > But why self-rag is worse than llama-3.1 in the original metric in terms of EM and F1?

---

> ### Author Response · Authors · 2024-12-01
> **Response to Dear Reviewer eSB7**
>
> Thank you for your response! We will provide further clarification on your concerns.
>
> # 1. Concerns on Self-RAG's performance in EM and F1
>
> **As far as we know, during data construction, Self-RAG evaluates the correctness of answers based on the reflection of a Critic model on the generated answers, without checking whether the output exactly matches the golden answer in the dataset. The Critic model is trained using data constructed by GPT-4 (as stated in the first paragraph of page 5 of the paper). As shown in the table below, GPT-4 tends to generate longer answers, meaning the output of GPT-4includes the answer but is not strictly identical to it:**
>
> ```
> +------------------------+---------------+---------------+---------------+---------------+
> |          Model         |       NQ      |    TriviaQA   |     PopQA     |     WebQA     |
> |                        +-------+-------+-------+-------+-------+-------+-------+-------+
> |                        |   EM  |  Acc  |   EM  |  Acc  |   F1  |  Acc  |   EM  |  Acc  |
> +------------------------+-------+-------+-------+-------+-------+-------+-------+-------+
> | gpt-4o-2024-08-06      | 16.9  | 54.0  | 69.1  | 82.1  | 48.2  | 55.7  | 15.1  | 55.2  |
> +------------------------+-------+-------+-------+-------+-------+-------+-------+-------+
> | gpt-4o-mini-2024-07-18 | 19.2  | 43.6  | 59.6  | 71.6  | 35.1  | 38.2  | 19.9  | 51.5  |
> +------------------------+-------+-------+-------+-------+-------+-------+-------+-------+
> ```
>
> **Therefore, GPT-4 may have a preference for longer answers, which may have some effect on selfrag's output distribution. This could also be one of the reasons why Self-RAG chooses Accuracy (Acc) as the evaluation metric.**
>
> In contrast, we use the golden answers provided by the dataset for data filtering, specifically determining whether the Final Answer is equivalent to the correct answer. This approach not only helps filter out hallucinations during the retrieval process but also positively impacts the output style of the answers. Self-RAG generates 4-20k data for each type of critic token (4 types) using GPT-4, while our method synthesizes 10k data based on open-source models, achieving better performance. Additionally, we use EM and F1 for evaluation to maintain consistency with FlashRAG, facilitating future comparisons with a broader range of work under the same setup.
>
>
> **Thank you again for your response. If our reply has addressed your concerns, we sincerely hope that you will reassess our work. We appreciate the time you have spent reviewing our response!**

---

> > ### Comment · Reviewer_eSB7 · 2024-12-03
> > **Response**
> >
> > Many thanks for the responses and experiments. In fact also for the last explanation and the possible weakness of selfrag.
> > I have revised my score, my remaining concern is about the novelty.
> > Thanks,

---

> > > ### Author Response · Authors · 2024-12-03
> > > **Thanks to Reviewer eSB7!**
> > >
> > > **Thank you for your hard work during the review and rebuttal process. We are pleased to have resolved your concerns. We also appreciate your final decision to increase the score!**

---

### Official Review · Reviewer_ZZb2 · 2024-11-03

**Soundness:** 3
**Presentation:** 3
**Contribution:** 3
**Rating:** 6
**Confidence:** 3

**Summary:**

This paper proposed a new method to automatically refine RAG retrieval query with reasoning. It showed that it can induce multi-step retrieval queries for multi-hop questions, and refine queries for simple questions, based on cases in Figure 1 and Figure 8. The authors fine-tuned Llama-3-Instruct-8B for the reasoning process, and used Qwen1.5-32B-Chat for crafting the query re-writing process. The performances of the new methods are better compared with other naive methods, single-time retrieval methods an iterative retrieval methods based on the results in the paper.

**Strengths:**

- The paper clearly described the methods with examples
- The paper employed recent open-source models like llama3-8b

**Weaknesses:**

- lack of comparison with GPT-4, GPT-4o or other closed AI models
- lack of case comparison with other RAG algorithms on multi-hop questions, in particular, for questions requiring a lot of reasoning process

**Questions:**

- Although the author provided reasoning examples in Figure 1, it did not compare with other RAG algorithms in case study section. Based on the case study part in Figure 8, the Auto-RAG refined the input query from "When was the first Ford F-150 produced" to "When was the first Ford F-150 model introduced". This is a relatively minor refinement on the query wording, instead of reasoning. If the authors can show more examples about reasoning in particular for multi-hop questions, comparing with other RAG methodologies, that would be great. The question in Figure 1 is an example multi-hop question. Some other similar questions can be found in relevant papers like "The Argentine PGA Championship record holder has won how many tournaments worldwide?" in the paper "Multi-hop Question Answering" https://arxiv.org/abs/2204.09140
- Can you also compare the performances with closed AI models, like GPT-4, GPT-4o etc.? For example, in the paper of "From Local to Global: A Graph RAG Approach to Query-Focused Summarization", it uses GPT-4 for the RAG solution.

---

> ### Author Response · Authors · 2024-11-19
> **Case Study with Multi-Hop Questions for Dear Reviewer ZZb2 [1/2]**
>
> Based on your valuable suggestions, we present a case study with multi-hop questions. We begin with the questions and their corresponding gold answers, followed by the outputs of Auto-RAG and other comparative baselines.
>
> **Question:** When did Edward Cromwell, 3Rd Baron Cromwell's father die?
>
> **Golden Answer:** 20 November 1592
>
> ## Auto-RAG:
>
> Step 0:
>
> Input:
> ```
> When did Edward Cromwell, 3Rd Baron Cromwell's father die?
> ```
>
> Output:
> ```
> Analysis: To answer this question, I would need to know the name of Edward Cromwell, 3rd Baron Cromwell's father, and his death date.
>
> Initial Query: Who is the father of Edward Cromwell, 3rd Baron Cromwell?
> ```
>
> Step 1:
>
> Input:
> ```
> Retrieved Document_1: Edward Cromwell, 3rd Baron Cromwell Edward Cromwell, 3rd Baron Cromwell (c. 1560 – 27 April 1607) was an English peer. He was the son of Henry Cromwell, 2nd Baron Cromwell by his wife Mary ......
> ```
>
> Output:
> ```
> Intermediate Answer_1: Based on the Retrieved Document_1, Edward Cromwell, 3rd Baron Cromwell's father is Henry Cromwell, 2nd Baron Cromwell. To answer the question, we need to find the date of death of Henry Cromwell, 2nd Baron Cromwell.
>
> Refined Query: When did Henry Cromwell, 2nd Baron Cromwell die?
> ```
>
> Step 2:
>
> Input:
> ```
> Retrieved Document_2: Ughtred, governor of Jersey, by whom she had two children: In 1537, three years after death of Anthony Ughtred in October 1534, she married Gregory Cromwell, the son of Thomas Cromwell and 1st Baron Cromwell ...... (Some irrelavant information)
> ```
> Output:
> ```
> Intermediate Answer_2: Based on the Retrieved Document_2, there is no information about the date of death of Henry Cromwell, 2nd Baron Cromwell. We need to refine our query and search again.
>
> Refined Query: What was the date of Henry Cromwell, 2nd Baron Cromwell's death?
> ```
>
> Step 3:
>
> Input:
> ```
> Retrieved Document_3: Cromwell Henry Cromwell, 2nd Baron Cromwell of Oakham (before 21 May 1538 – 20 November 1592) was an English peer......
> ```
> Output:
> ```
> Intermediate Answer_3: Based on the Retrieved Document_3, Henry Cromwell, 2nd Baron Cromwell died on 20 November 1592.
>
> Final Answer: 20 November 1592.
> ```
> ## FLARE
>
> Step 1:
>
> Input:
> ```
> <8-shot demonstrations with retrieved documents>
>
> Search results ranked based on relevance in descending order:
> [1]: Edward Cromwell, 3rd Baron Cromwell Edward Cromwell, 3rd Baron Cromwell (c. 1560 – 27 April 1607) was an English peer. He was the son of Henry Cromwell, 2nd Baron Cromwell by his wife Mary, daughter of John Paulet ...
> [2]: Down Cathedral, on 24 September 1607. Sir Arthur Chichester, when writing of his death to the council, 29 September 1607, expressed regret at his loss...
>
> Question: When did Edward Cromwell, 3Rd Baron Cromwell's father die?
>
> Answer:
> ```
> Output:
> ```
> Edward Cromwell, 3rd Baron Cromwell's father was Henry Cromwell, 2nd Baron Cromwell.
> ```
>
> Step 2:
>
> Input:
> ```
> <8-shot demonstrations with retrieved documents>
>
> Search results ranked based on relevance in descending order:
> [1]: presided over the Restoration of 1660. Richard Cromwell subsisted in straitened circumstances after his resignation......
> [2]: Oliver Cromwell Oliver Cromwell (25 April 15993 September 1658) was an English military and political leader......
>
> Question: When did Edward Cromwell, 3Rd Baron Cromwell's father die?
>
> Answer: Edward Cromwell, 3rd Baron Cromwell's father was Henry Cromwell, 2nd Baron Cromwell.
> ```
> Output:
> ```
> Henry Cromwell, 2nd Baron Cromwell died in 1658.
> ```
> Step 3:
>
> Input:
> ```
> <8-shot demonstrations with retrieved documents>
>
> Search results ranked based on relevance in descending order:
> [1]: Oliver Cromwell Oliver Cromwell (25 April 15993 September 1658)......
> [2]: presided over the Restoration of 1660. Richard Cromwell subsisted in straitened circumstances after his resignation ......
>
> Question: When did Edward Cromwell, 3Rd Baron Cromwell's father die?
>
> Answer: Edward Cromwell, 3rd Baron Cromwell's father was Henry Cromwell, 2nd Baron Cromwell. Henry Cromwell, 2nd Baron Cromwell died in 1658.
> ```
>
> Output:
> ```
> So the answer is 1658.
> ```

---

> ### Author Response · Authors · 2024-11-19
> **Case Study with Multi-Hop Questions for Dear Reviewer ZZb2 [2/2]**
>
> ## Self-RAG:
>
> Input:
> ```
> Question: When did Edward Cromwell, 3Rd Baron Cromwell's father die?
> ```
>
> Whether to retrieve?
>
> **True**
>
> Each Document with answers and reflection tokens:
>
> Doc1:
> ```
> Edward Cromwell, 3rd Baron Cromwell"\nEdward Cromwell, 3rd Baron Cromwell Edward Cromwell, 3rd Baron Cromwell (c. 1560 – 27 April 1607)......
> ```
> Answer 1:
> ```
> [Relevant]27 April 1607[Fully supported][Utility:5]
> ```
> Final Score 1: 3.88
>
> Doc 2:
> ```
> ......She died before 30 November 1631. Edward Cromwell’s son, Thomas visited his father’s former manor of Oakham in 1631 where he partook of the ancient tradition of forfeiting a horseshoe......
> ```
> Answer 2:
> ```
> [Relevant]1631[Fully supported][Utility:5]
> ```
> Final Score 2: 3.63
>
> Doc 3:
> ```
> "Richard Cromwell"\npresided over the Restoration of 1660. Richard Cromwell subsisted in straitened circumstances after his resignation......
> ```
> Answer 3:
> ```
> [Relevant]1658.[Fully supported][Utility:5]
> ```
> Final Score 3: 3.66
>
> Doc 4:
> ```
> "...... Edward Cromwell, 3rd Baron Cromwell Edward Cromwell, 3rd Baron Cromwell (c. 1560 – 27 April 1607) was an English peer......
> ```
> Answer 4:
> ```
> [Relevant]27 April 1607[Fully supported][Utility:5]
> ```
> **Final Score 4: 3.89**
>
> Doc5:
> ```
> "Edward Cromwell, 3rd Baron Cromwell"\nDown Cathedral, on 24 September 1607. Sir Arthur Chichester, when writing of his death to the council......
> ```
> Answer 5:
> ```
> [Relevant]1607[Fully supported][Utility:5]
> ```
> Final Score 5: 3.72
>
> Final Answer: 27 April 1607.

---

> ### Author Response · Authors · 2024-11-19
> **Response to Dear Reviewer ZZb2 [1/3]**
>
> **We sincerely thank the reviewer for the constructive and helpful feedback. We hope the following responses would help address your concerns.**
>
> # 1. Suggestions for additional examples of multi-hop questions.
>
> Thanks for your valuable suggestions!
>
>
> In our paper (Sec. 5.6 and Fig. 9), we conduct a case study to compare Auto-RAG and Self-RAG, clarifying their key **distinctions**. **Auto-RAG focuses on leveraging the reasoning and decision-making capabilities inherent in LLMs to perform iterative retrieval.** Auto-RAG determines the timing of retrieval based on the **relevance and availability of external knowledge**—continuing to retrieve if the information is irrelevant and proceeding to problem-solving when it is relevant. In contrast, Self-RAG's retrieval timing is dictated by the length of the generated text, with one retrieval triggered per sentence. Moreover, Auto-RAG articulates the rationale and process of retrieval in natural language, offering greater interpretability. In comparison, Self-RAG relies solely on reflection tokens for decision-making. Additionally, when encountering irrelevant information, Auto-RAG allows the model to decline answering and attempts further retrieval rather than forcing a response based on irrelevant data, thereby reducing hallucinations. On the other hand, Self-RAG compels the model to generate an answer for each retrieved document, regardless of its relevance.
>
> Due to space constraints, we present only a simplified case in the main text. Now, **we provide an additional example that involves multi-hop reasoning**, accompanied by a comparative analysis of our approach and other methods, to more clearly illustrate the advantages of our method. Here are the explanations:
>
> ### **Explanation for Auto-RAG:**
>
> For a complex multi-hop reasoning question, "When did Edward Cromwell, 3rd Baron Cromwell's father die?", the process unfolds as follows:
>
> Step 0: Auto-RAG **begins by conducting retrieval planning, identifying the necessary pieces of information**: (1) the identity of Edward Cromwell's father and (2) the time of his death. It then generates an initial query and decomposes the question into sub-questions, starting with: "Who is Edward Cromwell's father?"
>
> Step 1: From the retrieval results, **Auto-RAG successfully identifies Edward Cromwell's father and formulates a new, more specific query**: "When did Henry Cromwell, 2nd Baron Cromwell die?"
>
> Step 2: **Auto-RAG observes that the retrieved documents lack the required information. Rather than fabricating an answer based on irrelevant documents, it opts to slightly adjust the query, ensuring it remains aligned with the task.**
>
> Step 3: **Auto-RAG successfully retrieves relevant documents, finds sufficient information, and terminates the iterative retrieval process, producing the final answer.**
>
> ### **Explanation for FLARE:**
>
> In the first step, FLARE successfully identified Edward Cromwell's father. However, in the second step, due to the retrieval of irrelevant documents, FLARE generated hallucinatory responses. As a result, the third step produced an incorrect conclusion. Below are explanations of the characteristics of the FLARE method:
> * **High inference overhead.** FLARE employs few-shot prompting to facilitate multi-turn retrieval. The standard configuration utilizes 8-shot prompting, where each demonstration comprises two documents, one question, and a chain-of-thought response. While this setup effectively guides the model in reasoning on complex questions, it incurs significant computational overhead and increases the risk of generating hallucinations. *Auto-RAG can autonomously manage the retrieval process, achieving lower costs.*
> * **Unable to refuse to answer.** In the second step, due to the irrelevance of the retrieved documents, the model should have declined to provide an answer. Instead, the presence of few-shot demonstrations compelled the model to imitate the provided examples and produce a forced response, resulting in the model copying an unrelated date from the documents. *Auto-RAG is capable of rejecting irrelevant knowledge when answering questions, mitigating hallucination issues.*
> * **The retrieval strategy is not sufficiently flexible**. FLARE determines whether to refine its output based on the probability distribution of its responses. Nonetheless, irrelevant documents increase the likelihood of hallucinatory outputs, undermining the model's judgment. Consequently, FLARE ultimately generated a hallucinated response. *Auto-RAG employs natural language to articulate its reasoning and decision-making process, resulting in more precise decisions, enhanced interpretability, and better overall performance.*

---

> ### Author Response · Authors · 2024-11-19
> **Response to Dear Reviewer ZZb2 [2/3]**
>
> ### **Explanation for Self-RAG:**
>
> The core idea of Self-RAG is to independently generate responses based on multiple retrieved documents and reflect on their relevance through a reflection token, assessing whether the documents support the answer and the answer's overall utility. First, Self-RAG determines whether retrieval is necessary based on the input question. Then, for each document, Self-RAG generates a response and performs reflection, scoring each path based on the probability of extracting the reflection token. Finally, it selects the highest-scoring answer as the final result. The following are the differences between Self-RAG and Auto-RAG:
> * **Self-RAG generates a response for each document, regardless of its relevance.** Generating answers for all documents and selecting the most confident one as the final response may seem reasonable. However, this approach introduces unnecessary overhead and overlooks the relevance between documents. *Auto-RAG rejects irrelevant information, resulting in higher efficiency.*
> * **The retrieval strategy of Self-RAG is suboptimal.** Self-RAG’s retrieval strategy alternates between retrieval and generation. However, when all retrieved documents are irrelevant, the model is forced to generate an answer, leading to hallucinated outputs. Subsequently, the model has no opportunity to correct the generated content, resulting in error accumulation. In contrast, *Auto-RAG is more flexible in its generation timing, depending on the availability of external knowledge. When external knowledge is unavailable, it continues retrieval rather than forcing a generation, thereby mitigating hallucination issues.*
>
> ### Summary:
> **Auto-RAG focuses on leveraging the inherent reasoning and decision-making capabilities of LLMs for iterative retrieval. Auto-RAG autonomously adjusts its retrieval strategy based on the complexity of the question and the availability of external knowledge, achieving improved results.**
>
>
> **Thank you once again for your constructive feedback.** *Given the page limitations, we will include this section in Appendix C8, Page 28-31 to better assist youin understanding the innovation and effectiveness of our method.*
>
> **If my response has addressed your concerns, we kindly ask that you reassess our work. We would greatly appreciate it if you could consider raising your rating!**

---

> ### Author Response · Authors · 2024-11-19
> **Response to Dear Reviewer ZZb2 [3/3]**
>
> # 2. Suggestions for Supplementing GPT-4 Results
>
> Thanks for your valuable suggestions!
>
> Due to budget and time constraints, we sampled 1,000 samples from each dataset and compared the performance of our method with that of closed-source models. The random seed was set to 0.
> ```
> +--------------+------------------------+-------+-------+----------+-------+----------+-------+-------+
> | Method       | Model                  |   NQ  | 2Wiki | TriviaQA | PopQA | HotpotQA | WebQA | AVG   |
> +--------------+------------------------+-------+-------+----------+-------+----------+-------+-------+
> |                                             No Retrieval                                            |
> +--------------+------------------------+-------+-------+----------+-------+----------+-------+-------+
> | Naive-Gen    | gpt-4o-2024-08-06      |  16.9 | 43.2  |   69.1   | 48.2  |   48.3   | 15.1  | 40.1  |
> |              +------------------------+-------+-------+----------+-------+----------+-------+-------+
> |              | gpt-4o-mini-2024-07-18 |  19.2 | 31.7  |   59.6   | 35.1  |   37.9   | 19.9  | 33.9  |
> |              +------------------------+-------+-------+----------+-------+----------+-------+-------+
> |              | Llama-3-8B-Instruct    |  20.9 | 25.7  |   54.0   | 26.3  |   27.1   | 20.1  | 29.0  |
> +--------------+------------------------+-------+-------+----------+-------+----------+-------+-------+
> |                                          Standard Retrieval                                         |
> +--------------+------------------------+-------+-------+----------+-------+----------+-------+-------+
> | Standard RAG | gpt-4o-2024-08-06      |  14.0 | 36.2  |   58.7   | 45.6  |   46.8   | 13.9  | 35.9  |
> |              +------------------------+-------+-------+----------+-------+----------+-------+-------+
> |              | gpt-4o-mini-2024-07-18 |  29.9 | 34.0  |   61.3   | 49.6  |   45.6   | 19.5  | 40.0  |
> |              +------------------------+-------+-------+----------+-------+----------+-------+-------+
> |              | Llama-3-8B-Instruct    |  35.1 | 19.1  |   56.9   | 47.5  |   35.6   | 16.3  | 35.1  |
> +--------------+------------------------+-------+-------+----------+-------+----------+-------+-------+
> |                                         Autonomous Retrieval                                        |
> +--------------+------------------------+-------+-------+----------+-------+----------+-------+-------+
> | Auto-RAG     | Llama-3-8B-Instruct    | 34.2  | 47.9  |   58.6   | 48.4  |   45.7   | 23.4  | 43.0  |
> +--------------+------------------------+-------+-------+----------+-------+----------+-------+-------+
> ```
>
>
> The following presents the conclusions of the experimental results:
>
> 1. The average performance of Auto-RAG is the best.
> 2. GPT-4o demonstrated better performance without retrieval, while GPT-4o-mini showed improved performance after retrieval.
> 3. For a well-trained model, the quality of its parametric knowledge may be higher than that of external knowledge. Therefore, providing external knowledge may degrade its performance. Enhancing the model's ability to resist irrelevant information is crucial.
>
> **Auto-RAG autonomously adjusts its retrieval strategy based on the availability of external knowledge. When external knowledge is useful, it answers sub-questions, generates new queries, or derives the final conclusion. If the external knowledge is not useful, it refuses to answer and re-initiates the search process.**
>
> **Thank you once again for your constructive feedback.** *Due to space limitations and the fact that the test is based on a sample of 1,000 data points, which differs from the main experiment, we will provide additional details in Appendix A2, Page 17.*
>
> **If my response has addressed your concerns, we kindly ask that you reassess our work.**
>
> **We would greatly appreciate it if you could consider raising your rating!**

---

> > ### Comment · Reviewer_ZZb2 · 2024-12-01
> >
> > Thanks for adding the additional analysis.

---

### Author Response · Authors · 2024-11-22
**Kind Reminder for Submission13930 Reviewers**

Dear Submission13930 Reviewers:

We sincerely thank all reviewers for their constructive feedback during the review process! We have responded to your comments and made the corresponding revisions to the paper. If you have any further questions, please feel free to contact us, and we will respond promptly.

Best regards,

Submission13930 Authors

---

### Author Response · Authors · 2024-11-25
**Thank you to all the reviewers for your valuable feedback and time!**

Dear reviewers:

We sincerely thank all the reviewers for their diligent efforts during the review process. Your feedback has been instrumental in further enhancing the quality of our paper. Based on your valuable suggestions, we have revised the manuscript, and the key updates can be summarized as follows:
- Added experimental results comparing our approach with closed-source models (Appendix A2, Page 18).
- Investigated the impact of the order in which parametric knowledge and external knowledge are provided to Auto-RAG (Appendix A3, Pages 18–19).
- Included additional experiments comparing our approach with Self-RAG (Appendix A4, Page 19).
- Conducted additional case studies to compare the strengths and weaknesses of Auto-RAG with other iterative retrieval methods (Appendix C8, Pages 29–32).
- Updated Algorithm 1 and revised some detailed descriptions (Pages 3–4, highlighted in red).

We hope that our responses and revisions have addressed the your concerns. We would greatly appreciate it if you could take the time to review our replies in detail and consider raising your scores where appropriate. As the rebuttal period is nearing its end, please feel free to reach out if you have any further questions, and we will be more than happy to continue the discussion.

Best regards,

Submission 13930 Authors

---

### Comment · Area_Chair_r9LE · 2024-11-25
**Action Required: Respond to Author Rebuttals - Nov 27**

Dear ICLR Reviewers,

The author discussion phase is ending soon. Please promptly review and respond to author rebuttals for your assigned papers. Your engagement is critical for the decision-making process.

Deadlines:
- November 26: Last day for reviewers to ask questions to authors.
- November 27: Last day for authors to respond to reviewers.
- November 28 - December 10: Reviewer and area chair discussion phase.

Thank you for your timely attention to this matter.

---

### Author Response · Authors · 2024-11-27
**Kind Reminder for Submission13930 Reviewers**

Dear Submission13930 Reviewers:

We sincerely thank all reviewers for their constructive feedback during the review process!

**We responded to your reviews and revised the paper based on your feedback. We are unsure if our response has addressed all of your concerns.** If you still have any questions, we would be happy to continue the discussion with you.

Best regards,

Submission13930 Authors

---

### Meta-Review · Area_Chair_r9LE · 2024-12-21

**Metareview:**

The paper presents Auto-RAG, an autonomous iterative retrieval model that leverages LLMs' decision-making capabilities to enhance retrieval-augmented generation through multi-turn dialogues with the retriever. The approach involves synthesizing reasoning-based decision-making instructions and fine-tuning open-source LLMs. Reviewers acknowledge several strengths of the work, including comprehensive experimentation across six benchmarks, clear presentation, and practical value in addressing LLMs' context window limitations. The experimental results demonstrate performance improvements and the model's ability to autonomously adjust iteration counts based on question difficulty. However, reviewers raise some concerns about the work's contributions. First, the technical novelty appears limited, as automated iteration in prompt-based systems has been explored in previous works like ReACT, and the fine-tuning approach resembles existing methods. Second, there are questions about missing critical baselines (e.g., BeamAGGR, Self-CRAG) and some inconsistencies in reported results compared to existing work. Third, several technical details remain unclear, including the role of parametric knowledge, the inference pipeline's reliance on specific keywords, and training process specifics. While the authors have addressed some concerns through additional analysis and clarifications, particularly around baseline comparisons and experimental results, the fundamental questions about novelty and technical advancement remain. Overall, the work would benefit from substantial improvements before meeting the publication threshold.

**Additional Comments On Reviewer Discussion:**

Check the meta review.

---

### Decision · Program_Chairs · 2025-01-22

Reject